# Liquid–liquid phase separation within fibrillar networks

Jason X. Liu[1,2], Mikko P. Haataja [1,2], Andrej Košmrlj [1,2], Sujit S. Datta [3], Craig B. Arnold [1,2] & Rodney D. Priestley [2,3] ✉

Complex fibrillar networks mediate liquid–liquid phase separation of biomolecular condensates within the cell. Mechanical interactions between these condensates and the surrounding networks are increasingly implicated in the physiology of the condensates and yet, the physical principles underlying phase separation within intracellular media remain poorly understood. Here, we elucidate the dynamics and mechanics of liquid–liquid phase separation within fibrillar networks by condensing oil droplets within biopolymer gels. We find that condensates constrained within the network pore space grow in abrupt temporal bursts. The subsequent restructuring of condensates and concomitant network deformation is contingent on the fracture of network fibrils, which is determined by a competition between condensate capillarity and network strength. As a synthetic analog to intracellular phase separation, these results further our understanding of the mechanical interactions between biomolecular condensates and fibrillar networks in the cell.

Liquid–liquid phase separation plays a crucial role in living and soft matter systems. From the demixing of polymer blends to the growth of protein- and RNA-rich droplets known as biomolecular condensates, liquid–liquid phase separation yields material structures that serve a diverse array of functions[1,2]. The morphology and dynamics of phase separation are determined by a competition between the driving force for demixing, interfacial tension, and material mobility[3–7]. However, this picture is altered when phase separation occurs within a solid porous medium due to the influence of confinement, wetting, and material elasticity[8,9].

In the cell, biomolecular condensates are situated within a dynamic environment scaffolded by fibrillar networks of semiflexible polymers such as chromatin, F-actin, and microtubules[10,11]. Condensates residing in these environments exhibit physical behavior such as subdiffusion[12], suppressed coalescence[13–16], and aspherical morphologies[17,18]. While our understanding of physiological mechanisms involving biomolecular condensates has seen rapid progress in recent years[19,20], the physical picture of intracellular phase separation remains unclear due to the mechanical complexity of living biopolymer networks[10,21].

Recent experiments using rubbery polymer gels have begun to investigate the mechanical interactions between condensates and elastic media, demonstrating for example that rubbery polymer gels can arrest phase separation[22,23] and enhance material transport down stiffness gradients[24]. Phase separation within rubbery networks has additionally been demonstrated as an inspiration for materials design[25], for example by stiffening solids with liquid inclusions[26] or by generating structural color[23,27]. However, while these systems exhibit a rich phenomenology, the substantial size disparity between condensates (~µm) and molecular network strands (~nm) precludes direct interactions between condensates and individual network elements[28,29]. Rather, condensates necessarily probe only the bulk mechanical properties of these gels.

In contrast to rubbery polymer gels, semiflexible polymers can form fibrillar networks with mesh sizes ranging from hundreds of nm to several µm[30]. A condensate growing within a gel or network experiences no mechanical constraints until it becomes commensurate with the mesh size. However, when a condensate which does not wet the network fibrils grows beyond the mesh size, the condensate is forced to invade the network pore space and becomes mechanically

[1]Department of Mechanical and Aerospace Engineering, Princeton University, Princeton, NJ 08544, USA. [2]Princeton Materials Institute, Princeton University, Princeton, NJ 08544, USA. [3]Department of Chemical and Biological Engineering, Princeton University, Princeton, NJ 08544, USA. ✉e-mail: rpriestl@princeton.edu

constrained by fibrils to adopt a highly aspherical shape. Naturally, capillary forces act to minimize the condensate surface area, and thus there exists a competition between forces arising from condensate capillarity and the strength of restraining network elements. Unlike rubbery polymer gels which deform elastically, fibrillar networks such as those in the cellular interior readily undergo plastic deformation[11,31], and an understanding of the interplay between phase separation, condensate capillarity, and fibrillar network mechanics is crucial to elucidating the physics of intracellular phase separation.

Here, we report on the dynamics and mechanics of liquid–liquid phase separation within fibrillar networks. We induce phase separation within fibrillar biopolymer gels, where the large mesh size allows for direct observation of the phase separation kinetics and mechanical interactions between condensates and network structural elements. Experimentally, we first soak agarose hydrogels in ethanol to diffusively exchange water for ethanol. Next, we soak the ethanol-filled gels in solutions of 4% v/v decane in ethanol to yield gels which are saturated with ethanol-decane mixtures (Fig. 1a, i). We induce phase separation by subsequently soaking the mixture-loaded gels in water: as ethanol diffuses out of and water diffuses back into the gel, decane solubility within the mixture decreases. Eventually, the decane-ethanol-water mixture reaches an unstable composition which demixes into decane-rich and decane-poor phases (Fig. 1a, ii) (see "Methods" and Supplementary Note 1 for additional details). We find that growth of decane oil condensates confined within the tortuous, interconnected pore space proceeds via abrupt temporal bursts (Fig. 1a, iii). The commensurate size of oil condensates and the pore space enables visualization of direct mechanical interactions between

condensates and network elements, and our observations reveal that individual fibrils constrain the shape of condensates. By tuning the competition between condensate capillarity and network strength, we show that capillary forces can fracture restraining fibrillar elements and plastically deform the network (Fig. 1a, iv–v).

## Results

### Phase separation within fibrillar networks

We employ fluorescence confocal microscopy to directly observe the kinetics and mechanics of liquid–liquid phase separation within fibrillar agarose networks. Fig. 1b and Supplementary Movie 1 show a time series of an individual decane oil condensate (visible in the bright-field and red fluorescence channels) growing within a 0.8% w/w agarose gel with a mesh size of approximately $\xi \approx 1.3\,\mu m$ (agarose visible in the green fluorescence channel; mesh sizes in Supplementary Note 2). Temporally tracking the in-plane area of the condensate in the bright-field images of Fig. 1b reveals that condensate growth occurs via abrupt jumps in area which are separated by quiescent periods (Fig. 1c). These abrupt increases in the in-plane area are clearly visible in Fig. 1c at $t = 26\,s$ and $66\,s$, corresponding to the regions in Figs. 1b$_2$ and b$_3$ marked I and II. Region I appears within a single 2-second frame, and region II appears as a series of three sequential jumps, each of which occurs within a 2-second frame. Phase separation initiates when the ethanol-water-decane mixture reaches an unstable composition (Supplementary Fig. 1) and proceeds until the oil solute is fully depleted from solution. In Fig. 1d and Supplementary Movies 3 and 4, this occurs by about $t = 500\,s$, when no further growth of any condensates is observed in the full field-of-view.

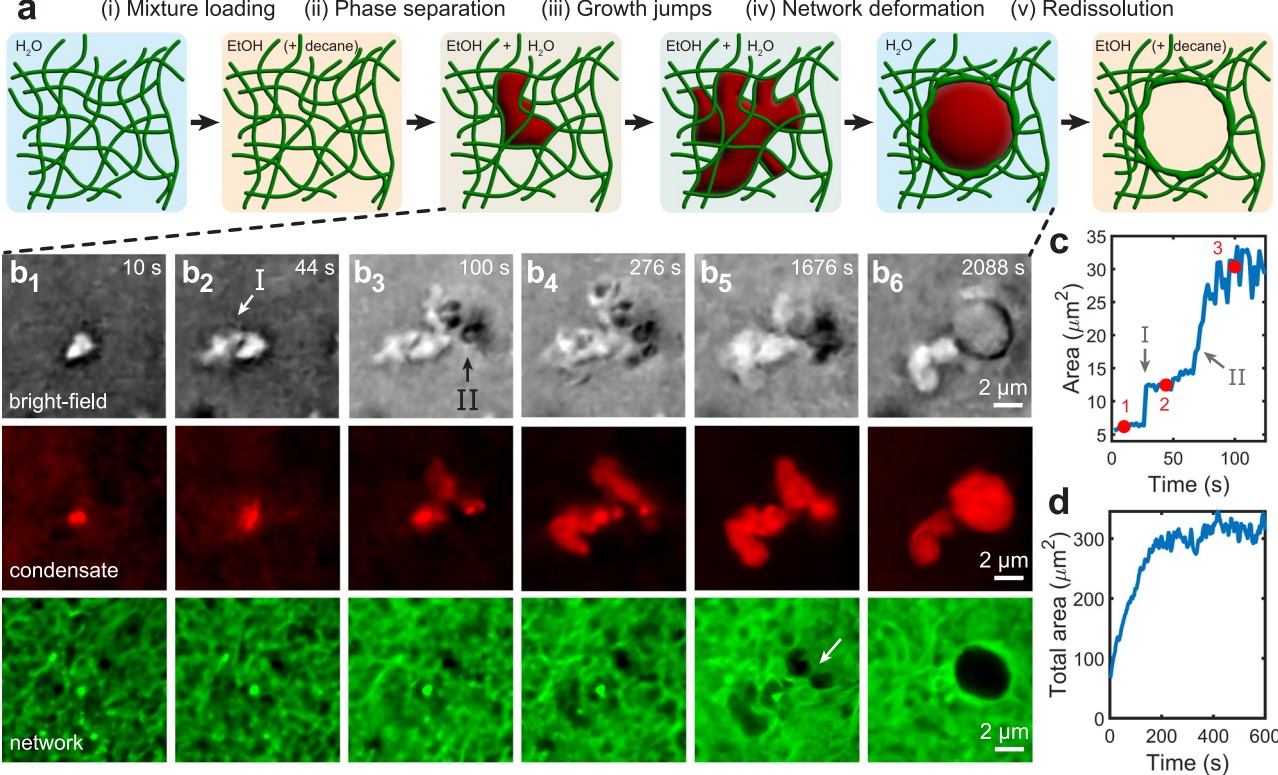

**Fig. 1 | Kinetics of liquid–liquid phase separation within fibrillar networks. a** Schematic depicting the phase separation of oil condensates within a fibrillar network. Blue represents water, tan represents ethanol, green represents agarose fibrils, and red represents decane. **b** Bright-field and fluorescence confocal microscopy time series showing the condensation of decane (bright-field and red) within a 0.8% w/w agarose network (green) (Supplementary Movies 1 and 2). Growth of the condensate within the fibrillar network occurs between (b$_{1–4}$), followed by network deformation (b$_5$), and finally fracture of the restraining element indicated by the arrow in (b$_5$), which leads to fluid rearrangement and expansion of a cavity within the network (b$_6$). **c** Evolution of the condensate in-plane area vs. time, where the in-plane area is determined from the bright-field images. Numbered red circles correspond to panels (b$_{1–3}$). Abrupt jumps are observed at $t = 26$ and $66\,s$, at numerals I and II. The corresponding regions are marked in (b$_2$) and (b$_3$). **d** Total in-plane area of all condensates in the full field-of-view movie as a function of time (Supplementary Movies 3 and 4). Area growth ceases by approximately $t = 500\,s$, indicating that the oil solute has been fully depleted.

As the water content rises during solvent exchange, there is also a corresponding increase in the condensate interfacial tension, $\gamma_{ow}$ (Supplementary Fig. 4). Initially, the oil condensate is observed to adopt a highly tortuous morphology which permeates throughout the gel pore space without deforming the surrounding network (Fig. 1b$_{1-4}$). Condensate capillarity is yet insufficient to deform the surrounding network fibrils since the high ethanol content early in the solvent-exchange process maintains a low $\gamma_{ow}$ (Supplementary Fig. 5). However, $\gamma_{ow}$ continually rises, and by $t = 1676$ s, the condensate exhibits a more rounded shape in Fig. 1b$_5$ (compared to Fig. 1b$_4$) as capillary forces seek to minimize the condensate surface area. Concomitant deformation of the network fibrils immediately adjacent to the condensate is additionally visible in the green fluorescent image of Fig. 1b$_5$. Eventually, capillary forces become sufficient to fracture the restraining fibrillar element indicated by the white arrow in the green channel of Fig. 1b$_5$, thereby allowing the tortuous condensate to restructure into a spherical droplet (Fig. 1b$_6$) (Supplementary Movie 2).

These results suggest that there are two time scales which determine the relevant kinetics: that of phase separation and that of increasing $\gamma_{ow}$. Condensate growth can occur via abrupt jumps within the network pore space if phase separation occurs before the interfacial tension has risen sufficiently to deform the network. Alternatively, if capillary forces become large enough to deform the network prior to completion of phase separation, then it is possible for condensate growth and network deformation to be coupled.

## Phase separation via abrupt jumps

Growth via abrupt jumps is also observed in gels of different mesh sizes, with condensates increasing in tortuosity as they permeate the yet smaller pore spaces attained with increasing gel concentration (Supplementary Note 3 and Supplementary Movies 5–8). To elucidate the mechanisms which underlie the observed growth kinetics, in Fig. 2 and Supplementary Movie 9 we show a time series of a condensate undergoing growth jumps within a 0.3% w/w gel which has a larger mesh size of $\xi \approx 3.7$ µm, facilitating visualization. In Fig. 2c, we track the size of a pre-existing condensate lobe, quantified by the width of the condensate between the two blue markers in Fig. 2a$_1$; this reveals that when growth jumps occur at $t = 54$ s and 74 s, there are simultaneous, sudden reductions in the pre-existing lobe width. The volume flow rates associated with the appearance of regions I and II are at least 3.0

and 7.4 fL/s, respectively, which is substantially larger than the volume growth rate of a condensate which is unconstrained by the fibrillar network, 0.23 fL/s (Supplementary Note 3). These observations imply that regions I and II form via fluid redistribution from the main condensate body into an adjacent, empty pore space, rather than as a jump in the local condensation rate. Moreover, we observe that the evolving fluid shape bears directly on the local deformation of the network. This is demonstrated by the simultaneous shrinkage of the fibrillar cage which confines the pre-existing lobe during a growth jump: a reduction in the size of the green network cavity can be seen in Supplementary Movie 9 as fluid redistributes into regions I and II.

Since decane is a non-wetting phase for the agarose gel (Supplementary Note 4), these observations are consistent with a picture of capillarity-driven immiscible fluid flow within a solid porous matrix[32,33], where the growth of oil condensates throughout the fibrillar network is thermodynamically driven by solvent exchange[22]. As a condensate grows due to phase separation, the pressure it experiences increases through two mechanisms. Firstly, the fluid capillary pressure rises as the condensate's menisci advance towards narrower pore constrictions formed by the confining fibrils[33,34]. Secondly, the growing condensate expands the confining cage of network fibrils, storing elastic energy via network deformations[32]. When the meniscus of any one particular fluid lobe crosses a pore throat, capillary forces rapidly drive the fluid into the adjacent cavity, thus relaxing the capillary pressure as well as elastic network deformations. This process repeats so long as oil continues to condense from solution.

These growth kinetics are analogous to conventional Haines jumps in porous media, where sudden advances of a non-wetting, immiscible fluid during invasion are accompanied by reductions in the capillary pressure[35,36]. However, rather than having an externally applied pressure driving flow, in this system it is phase separation which drives these jumps, where the driving pressure is determined by the local oil supersaturation[22]. We additionally estimate that the capillary number associated with these growth kinetics is $Ca = 1.3 \times 10^{-9}$ (Supplementary Note 5), within the regime expected for capillarity-driven restructuring of the fluid interface, at $Ca \ll 1$[36].

## Fibrillar network deformation and fracture

Condensates possess an extended structure which permeates the network pore space, and forces supplied by network fibrils compete

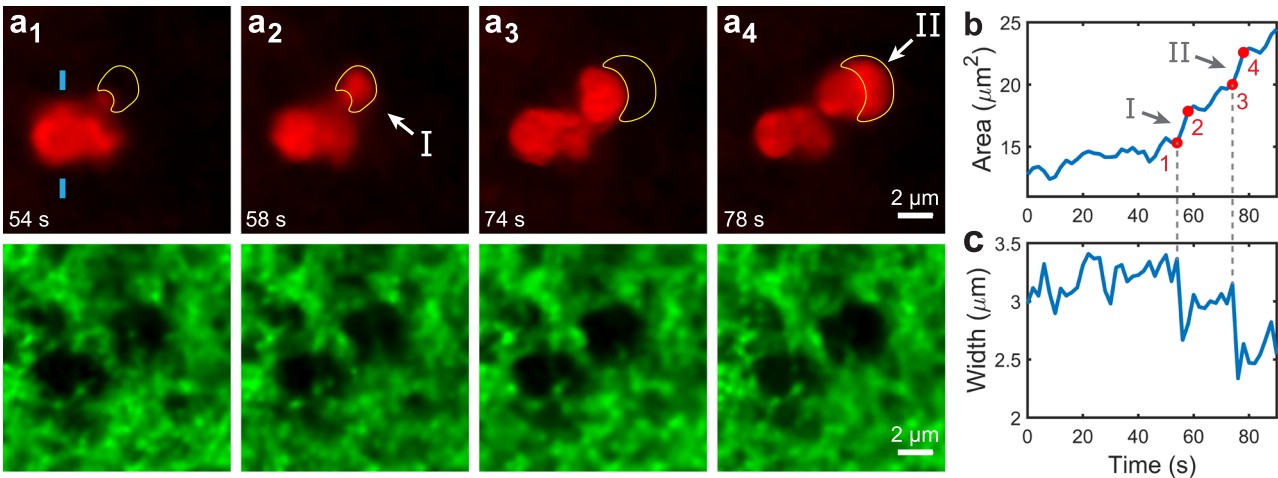

**Fig. 2 | Condensate growth via abrupt interface jumps. a** Fluorescence confocal microscopy time series showing abrupt growth jumps of a decane condensate (red) within a 0.3% w/w agarose network (green). The yellow shapes identify regions I and II which appear in abrupt interface jumps. **b** Evolution of condensate in-plane area vs. time. Numbered red circles correspond to (a$_{1-4}$). Abrupt growth jumps are observed at gray numerals I and II, corresponding to the regions marked in (a$_2$) and (a$_4$). **c** Width of the condensate lobe between the blue markers of (a$_1$), plotted vs. time. Sudden reductions in the lobe width are experienced at $t = 54$ and 74 s. These reductions occur concurrently with the growth jumps in (**b**), indicating that regions I and II form via fluid redistribution from pre-existing lobes. Vertical dashed lines between (**b**) and (**c**) are guides to the eye.

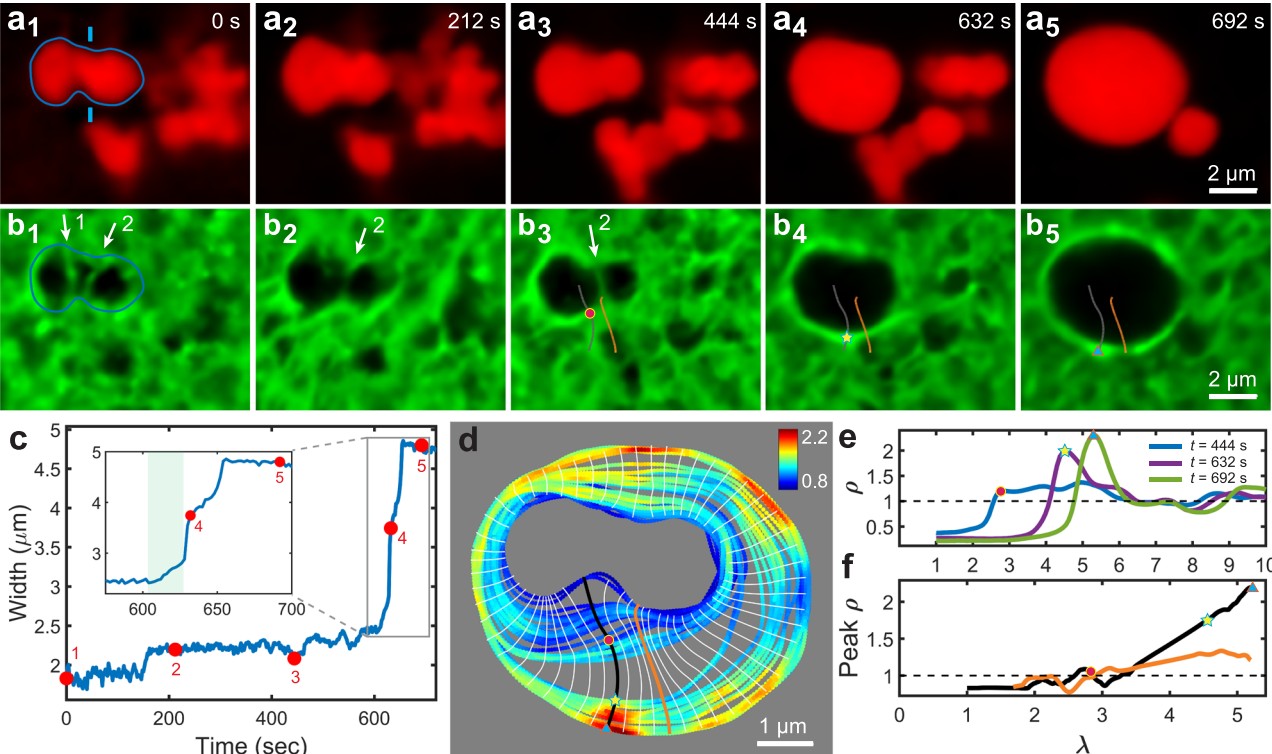

**Fig. 3 | Fibril fracture and network compaction by restructuring condensates.**
**a**, **b** Confocal fluorescence microscopy time series of a decane condensate (red) restructuring into a roughly spherical droplet after fracturing restraining fibrils in a 0.8% w/w agarose network (green). Numbered white arrows in (b$_{1-3}$) indicate the individual network elements which restrain the condensate. **c** Time evolution of the condensate width as measured between the blue markers in (a$_1$). Numbered red circles correspond to panels (a$_{1-5}$). At $t = 152$ s, the left restraining element in (b$_1$) fractures, allowing for the condensate to expand slightly. At $t = 628$ s, restraining element 2 fractures, allowing the tortuous condensate to minimize its surface area by restructuring into a roughly spherical droplet. The inset shows a zoom-in to the fracture event in which fibril elongation is highlighted in green. **d** Profiles of the edge of the oil droplet at different times throughout the cavity formation process

(example profile shown in blue in a$_1$). Profiles are colored according to the intensity of the network fluorescence along the profile, normalized to the mean background intensity. The white curves, perpendicular to the colored profiles, are expansion trajectories. **e** Spatial profile of the relative network material density, $\rho$, plotted along the black trajectory at $t = 444$, 632, and 692 s (corresponding to b$_{3-5}$). The network fluorescence intensity is used as a proxy for $\rho$ and is normalized to the mean background intensity. The distance along the trajectory, $\lambda$, is normalized to the mesh radius, $\xi/2 = 0.65$ μm. The circle, star, and triangle mark the peak density and correspond to the markers in (b$_{3-5}$, **d**, and **f**). **f** Peak density extracted from (**d**) as a function of $\lambda$ along the black and orange expansion trajectories, depicting the rise in peak $\rho$ as the restructuring condensate drives the cavity to expand and compactifies the surrounding network.

against condensate capillarity to stabilize these highly aspherical bodies. Unlike the polymer strands in rubbery gels, whose mechanical response largely stems from entropic forces[37], the mechanical response of individual semiflexible polymer fibrils such as agarose is largely enthalpic. As a result, the bending stiffness and tensile modulus of the constituent fibrils dictate the mechanics of fibrillar networks[38,39]. Due to the low connectivity of fibrillar agarose networks, the initial mechanical response to capillary forces primarily arises from bending, as opposed to stretching, of the individual fibrils[40].

At small strains, these bending deformations are elastic, as evident from the relaxation of network cavities which occurs during the fluid redistribution events shown in Fig. 2. As solvent exchange proceeds, $\gamma_{ow}$ continually rises, and as increasing capillary forces act upon the surrounding network elements, fibrils begin to yield plastically by bending to allow the condensate to reduce its surface area, such as visible between Fig. 1b$_4$ and 1b$_5$. However, these deformations become arrested near the mesh size due to topological constraints: deformation solely via fibril bending is not possible far beyond the mesh size because the fibrillar strands of the network consist of closed loops which constrain the condensate. The network stiffens considerably as constraining fibrils align in the load-bearing direction and the mechanical response transitions from fibril bending to stretching;[41] in bulk agarose gels, nearly an order-of-magnitude increase in the shear modulus is experienced during this transition[40,42]. In this fibril stretching-dominated regime, stresses are heterogeneously

distributed throughout the network, with forces concentrated within the specific elements which constrain the condensate[43].

Figure 3a, b and Supplementary Movie 10 depict a condensate within a 0.8% w/w agarose gel after the oil solute has fully phase separated. Individual fibrillar elements which constrain the condensate are visible in Fig. 3b$_{1-3}$, numbered with white arrows. These load-bearing network elements fail when their tensile strength is exceeded, and the condensate expands slightly at $t = 152$ s when element 1 fractures. Capillary forces are subsequently transferred to element 2, which eventually fails at $t = 628$ s. We temporally track the oil condensate width between the blue markers in Fig. 3a$_1$ as a proxy for the length of restraining element 2, and this reveals mesoscopic fibril elongation prior to fracture (green highlighted region in the inset of Fig. 3c). Direct observations reveal the thinning of agarose fibrils upon stretching (Supplementary Movie 11), suggesting a confluence of fibril elongation mechanisms including partial fracture and relative sliding of constituent agarose polymer chains[44,45]. By assuming that element 2 forms a circular hoop around the condensate, we estimate an agarose fibril tensile strength of ~340 MPa (Supplementary Note 6).

Upon network fracture, there is a sudden reduction in the local forces restraining the condensate, and condensate capillarity drives fluid flow into the nascent cavity, buckling and compacting the adjacent network fibrils as the cavity grows beyond the mesh size (Fig. 3a$_{4-5}$). Thus, cavity formation is primarily dependent on the fracture of individual restraining fibrillar elements. This is different

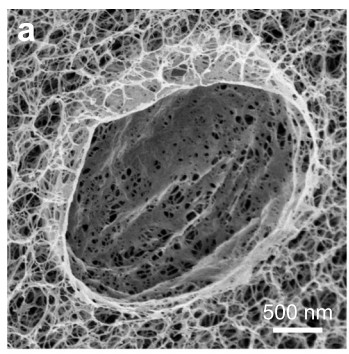
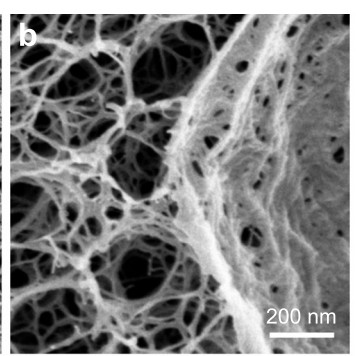
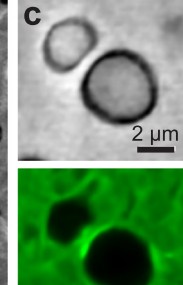

**Fig. 4 | Structure of the densified network around restructured condensates. a, b** Cross-sectional SEM images of a 2.0% w/w agarose gel after oil phase separation, network fracture, and cavity expansion. The sample is frozen in liquid-nitrogen-cooled liquid ethane to circumvent freezing artifacts, and the oil and aqueous phases are subsequently removed by lyophilization. **c** Bright-field and confocal fluorescence microscopy images showing the densified network (green) precluding the coalescence of two adjacent oil droplets (bright-field) in a 1.3% w/w gel.

from cavity formation in rubbery polymer gels, where condensate expansion is counteracted by hyperelastic deformation of the surrounding network strands[29,46,47]. In such a rubbery elastic gel, a cavity will grow upon the subjection of a driving pressure which exceeds a critical value of $p_c = 5E/6$[48], where $E$ is the Young's modulus. These mechanistic differences in cavity formation arise from the different mechanical responses of rubbery gels versus fibrillar networks. While bulk rubber elasticity counteracts cavity expansion in a rubbery polymer gel, the local fracture mechanics of the restraining fibrils dictates cavity growth in a fibrillar network.

## Cavity growth and network compaction

Since the condensate size is commensurate with the network mesh size, tortuous condensates explore local variations in the network pore size, connectivity, and density, thus experiencing variable mechanical resistance from the network[39]. This spatially variable resistance, together with local variations in the magnitude of capillarity forces, results in anisotropic cavity expansion upon fracture. Fig. 3d shows overlaid condensate shape profiles at various times during cavity growth where, for example, the cavity expands over 3 μm in the downwards direction, but less than 1 μm leftwards (also see Supplementary Movie 10). We additionally observe multistep cavity growth (inset of 3c), which suggests that during cavity expansion, stress is successively redistributed onto new elements which may temporarily restrain the droplet. This result stands in contrast to the isotropic droplet growth observed in rubbery polymer gels, where the mechanical resistance is spatially homogeneous[22,29].

As the cavity expands beyond the mesh size, the surrounding network is compacted into a shell whose material density, $\rho$, increases with deformation extent. Using the network fluorescence brightness as a proxy for $\rho$, in Fig. 3e, we plot spatial profiles of $\rho$ along the black trajectory of Fig. 3d at three different times. At $t = 444$ s, no substantial build-up in $\rho$ is observed yet at the edge of the cavity. However, as the restructuring condensate drives cavity expansion, more material is accumulated into this shell, and by $t = 632$ and 692 s, the shell exhibits a peak $\rho$ which is 2- and 2.3-times greater than the background density, respectively.

Each droplet edge profile in Fig. 3d is colored according to the local network $\rho$, revealing substantial azimuthal variations in network accumulation. We exemplify this by considering the rise in shell density as a function of displacement along the black and orange trajectories of Fig. 3d. In Fig. 3f, we can see that this shell peak $\rho$ rises to 2.1 times the background density along the black trajectory, whereas only a 1.3-times enhancement is experienced along the orange trajectory. In Fig. 3b₃, the black and orange trajectories are overlaid on an image of the network prior to fracture and cavity expansion, where it can be seen that the black trajectory overlays a more dense region whereas

the orange trajectory traverses a substantial void. Since material along the path of an expansion trajectory is accumulated into the shell as the cavity grows, we find that the difference in peak $\rho$ between the two trajectories directly reflects the heterogeneous material density distribution of the underlying network.

The extent of densification that occurs during cavity formation in fibrillar networks stands in contrast to that of rubbery polymer gels, whose networks do not accumulate substantially around growing droplets due to the rubbery gels' largely affine deformation[29,49]. This reflects the high compressibility of the fibrillar network, as the network can densify substantially through the buckling and compaction of fibrils. To visualize the deformed network microstructure, we rapidly freeze gels in liquid-nitrogen-cooled liquid ethane to avoid freezing artifacts, lyophilize the gels to remove the frozen water and oil phases, and view the fractured gel cross sections with scanning electron microscopy (SEM). The SEM images in Fig. 4a, b reveal a densely packed shell of agarose surrounding the former location of an oil droplet. The shell is approximately 100 nm thick for a 2.0% w/w gel, and its composition of many compacted individual fibrils is apparent in Fig. 4b. The SEM images additionally demonstrate that the deformation field is spatially limited; network fibrils immediately exterior to the densified shell (Fig. 4b) exhibit a morphology similar to that of the undeformed gel (Supplementary Fig. 22b), implying a highly localized deformation. We additionally find that this densified shell can form a barrier which prevents direct contact and coalescence between condensates grown adjacent to one another, such as the condensates shown in Fig. 4c.

While polymer gels can sustain high and reversible strains due to rubber elasticity[29], networks of semiflexible fibrils such as agarose deform plastically upon the application of large strains due to the fibrils' athermal nature[40,42]. Thus, while the deformation induced by liquid–liquid phase separation in rubbery gels is reversible[22,24,29,50], we find that phase separation permanently deforms agarose networks. To probe the reversibility of network deformation, we remove the phase separated oil phase from the fibrillar networks by covering gels with pure ethanol to dissolve away the oil, and in Fig. 5a we show an optical microscopy time series of this process in a 0.3% w/w gel (Supplementary Movie 12). We observe that cavities largely maintain their shape, with a 20% shrinkage in cavity area observed after oil dissolution and with similar shrinkage observed in higher concentration gels (Supplementary Note 7). When we reduce the condensate interfacial tension via addition of the surfactant laureth-4 at 5% v/v in all solvents and mixtures, we find that network fracture is precluded since condensate capillarity never rises sufficiently to fracture restraining fibrils (Fig. 5b). Yet, the condensate slightly deforms the network even under these conditions, and we observe remnant plastic deformation after oil dissolution in Fig. 5b.

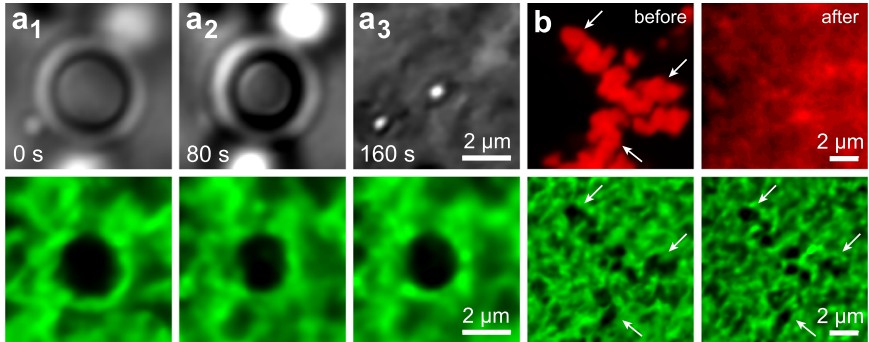

**Fig. 5 | Dissolution experiments demonstrate that network deformation is plastic. a** Bright-field and confocal fluorescence microscopy time series showing the dissolution of a decane condensate (bright-field) in a 0.3% w/w agarose gel (green). The network cavity in (a₃) has a 20% smaller area than the cavity in (a₁). No surfactant is present in this experiment. **b** Confocal fluorescence microscopy images depicting a decane condensate (red) in a 0.8% w/w agarose gel (green),

before and after dissolution. The surfactant laureth-4 has been added at a concentration of 5% v/v to reduce the oil–water interfacial tension, $\gamma_{ow}$, and prevent network fracture. The network is deformed slightly by the condensate, and this deformation persists after oil dissolution; white arrows are a guide to identify this mesh-scale network deformation.

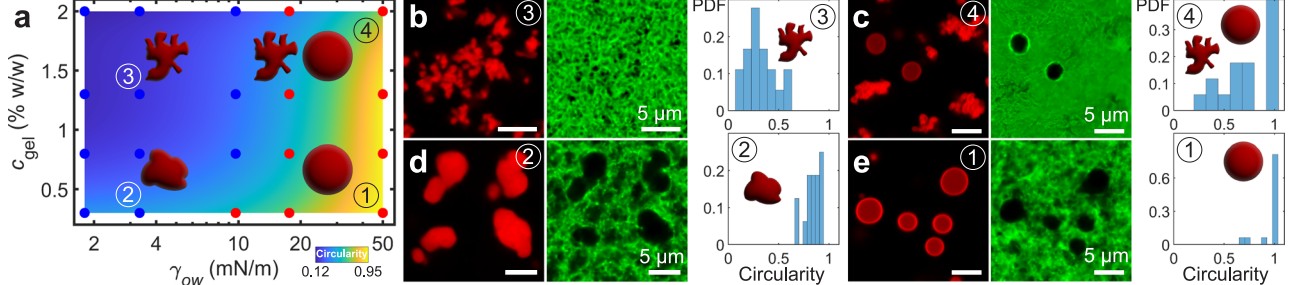

**Fig. 6 | Competition between condensate capillarity and network strength. a** Morphology state diagram of oil condensates as a function of gel concentration and interfacial tension, $\gamma_{ow}$, which is varied via addition of the surfactant Triton X-100. The diagram is colored according to the average circularity of condensates present in a given sample (colorbar below diagram). Red dots correspond to samples which exhibit network fracture, as inferred by the presence of any

condensates which possess a circularity above 0.95. The schematic shapes depict representative condensate morphologies which appear in the corresponding samples ①–④. **b–e** Confocal fluorescence micrographs of decane (red) and the agarose network (green), as well as probability distribution function histograms of the condensates' circularity for samples ①–④.

## Tuning capillarity and network strength

Finally, to investigate the competition between condensate capillarity and network strength, we form gels of varying agarose concentration and systematically vary the oil–water interfacial tension, $\gamma_{ow}$, via addition of the surfactant Triton X-100 (TX-100) (Supplementary Note 8). A reduction in $\gamma_{ow}$ precludes network fracture by reducing the magnitude of capillary forces exerted on fibrils and preserves the tortuous morphology of condensates, reflected by a corresponding reduction in average condensate circularity (Fig. 6a). We observe four typical morphologies, schematically depicted in Fig. 6a. In dilute gels ($c_{gel}$ = 0.3% w/w) with no added surfactant ($\gamma_{ow}$= 50 mN/m), ①, interfacial tension dominates and spherical droplets are exclusively observed in the final morphologies (Fig. 6e). Reducing the interfacial tension to 2 mN/m in ② yields aspherical condensates which exhibit deformed protrusions commensurate with the mesh size (Fig. 6d). Upon an increase of $c_{gel}$ to 1.3% w/w, condensates in ③ exhibit higher tortuosity and reduced circularity as they pervade yet narrower pore spaces (Fig. 6b). Network fracture is prevented in both ② and ③, although the fibrillar network still yields via fibril bending to accommodate the condensates. Interestingly, in a 2.0% w/w gel at $\gamma_{ow}$ = 50 mN/m, ④, we observe a bimodal population of spherical and mesh-constrained condensates (Fig. 6c). While a bimodal population has been predicted to arise kinetically during phase separation due to solute depletion in the vicinity of cavitated droplets[46], here, solute is already fully depleted prior to network fracture. Rather, we hypothesize that the bimodal population

reflects heterogeneities in the local fracture strength of the underlying network.

In dilute gels (0.3% w/w), condensates gradually become more spherical as interfacial tension is increased, illustrated by the monomodal circularity histograms whose average values increase with increasing $\gamma_{ow}$ (Fig. 6d, e and Supplementary Fig. 21). The lateral size of droplets is comparable to the mesh size in these gels, which allows the network to accommodate a greater extent of fluid restructuring through fibril bending prior to network fracture. In contrast, this transition is abrupt in more concentrated gels, with a population of spherical droplets emerging suddenly with increasing $\gamma_{ow}$ (Supplementary Fig. 20). Since the condensate size exceeds the mesh size by several times in higher concentration gels, deformation to intermediate morphologies is limited. Instead, condensates transition directly from mesh-constrained to spherical morphologies through fracture.

## Discussion

These experiments demonstrate several basic mechanisms of liquid–liquid phase separation within fibrillar networks: that non-wetting condensates can grow via abrupt capillarity-driven fluid restructuring events, that rearrangement of tortuous condensates into spherical droplets is contingent on fracture of restraining fibrillar elements, and that network deformation is highly plastic. The agarose networks in which we demonstrate these phenomena recapitulate several properties of the networks which scaffold the intracellular environment, such as the mesh size and athermal nature of the fibrils'

mechanical response[40,42]. However, intracellular networks exhibit additional complexities which are not captured here. For example, the crosslinks which form between agarose fibrils are permanent[40]; on the other hand, intracellular networks contain dynamic and reconfigurable crosslinks which enable cells to remodel their interiors in response to mechanical forces[11]. In the cytoplasm, actin binding proteins reversibly bind actin filaments together[51,52], and similarly in the nucleus, chromatin transiently binds with itself via chromatin binding proteins[52,53] as well as complex formation[54]. These dynamic crosslinks allow the cytoskeleton[55,56] and nucleoplasm[57,58] to respond elastically to mechanical forces on short time scales and be remodeled over longer times. In the context of intracellular phase separation, such a viscoelastic response would allow for stress relaxation during biomolecular condensate growth. Since condensate growth via abrupt jumps as well as network fracture is a direct consequence of a rise in the capillary pressure, manifestation of these phenomena within the cell would depend on a competition of time scales. If the time scale of network rearrangement is shorter than that of condensate growth, then fibrillar elements can accommodate non-wetting condensate growth and prevent the rise in capillary pressure that would arise due to growth into pore throat constrictions, thus precluding growth via abrupt interface jumps. Similarly, if network elements are able to restructure and relax stresses more quickly than condensate capillarity rises, this would allow networks to accommodate the restructuring of condensates into spherical droplets via crosslink reorganization rather than fibril fracture[59]. Furthermore, both the cytoskeleton[60,61] and nucleoplasm[62–65] are active environments in which motor proteins consume ATP to enable force production on fibrillar elements. These network-generated forces compete against condensate capillarity and can dynamically deform condensates by squeezing or even breaking them apart[66,67].

Additional complexities arise when considering the rheology of the condensates themselves. These experiments consider the phase separation of simple fluids. However, the composition of biomolecular condensates is substantially more complex, as condensates form via multivalent protein and nucleic acid interactions which drive phase separation[6,68–70]. These interactions yield condensates which exhibit viscoelastic properties that depend on conditions such as the salt concentration[71], age[72], or composition[73]. With increasing associative interactions among constituents, condensates exhibit slower internal dynamics and may achieve increasingly gel-like states[74], and upon gelation, condensates are more able to resist internal forces arising from condensate capillarity as well as external forces such as those applied by active fiber networks[10]. Lastly, while our experiments mimic biomolecular condensates which do not wet the networks they reside in, some condensates have attractive interactions with network fibrils. For example, tau protein condensates will wet microtubules[18,75,76], leading to capillary forces which pull on network elements and which can induce large-scale amplification of network stresses[77].

In conclusion, phase separation within fibrillar networks offers us an avenue to approach a host of fundamental and applied problems. The use of synthetic and reconstituted analogs to biomolecular condensates and fibrillar networks will provide us new opportunities to further our understanding of the mechanics of liquid–liquid phase separation in the cellular interior. Additionally, liquid–liquid phase separation within fibrillar networks offers a versatile design motif in the creation of solid–liquid composite materials. By extending our knowledge of classic phase separation phenomena into new contexts, these results offer us the possibility of deepening our understanding of the rules of life as well as a means to develop novel material architectures.

## Methods
### Materials
Type I-A, low electroendosmosis agarose was purchased from Sigma Aldrich. Deionized water was filtered through a 0.2 $\mu$m filter using a NANOpure Diamond filtration system prior to use. Anhydrous, 200 proof ethanol was purchased from Sigma Aldrich. $n$-decane (>99% purity) was purchased from TCI Chemicals. 5-([4,6-Dichlorotriazin-2-yl]amino)fluorescein hydrochloride (DTAF) (≥90% purity) was purchased from Sigma Aldrich. 1,6-Diphenyl-1,3,5-hexatriene (DPH) (>98% purity) was purchased from Sigma Aldrich. BDP 558/568 NHS ester (BDP 558) was purchased from Lumiprobe Corporation. Sodium sulfate (Na$_2$SO$_4$, >99% purity, anhydrous) was purchased from Sigma Aldrich. Sodium hydroxide (NaOH, >98% purity, anhydrous) was purchased from Sigma Aldrich. Triton X-100 (TX-100) was purchased from Sigma Aldrich. Laureth-4 (L4) was purchased from Sigma Aldrich.

### Preparation of agarose gels
Agarose powder was dissolved in boiling water at mass concentrations of 0.3%, 0.8%, 1.3%, and 2.0% w/w. During boiling, the agarose concentration was maintained by the continual addition of water to replenish that lost due to evaporation. Fully dissolved agarose solutions were subsequently pipetted into flat-bottomed petri dishes and allowed to cool at 25 °C for 2 h to form solid gels (0.4 mL gel volume).

### Solvent-exchange condensation
Liquid–liquid phase separation within agarose gels was achieved via solvent exchange. After gel formation, the solvent was first exchanged from water to ethanol by soaking the gels in 5 mL of ethanol for 12 h. Agarose gels are known to be stable and not to shrink significantly during exchange from water to ethanol. Subsequently, the ethanol was decanted and the ethanol-filled gels were soaked for 12 h in a 5 mL mixture of decane in ethanol (4% v/v), with a hydrophobic fluorescent dye co-dissolved (details in "Fluorescent labeling of decane"). In surfactant experiments, the nonionic surfactant Triton X-100 was co-dissolved in ethanol as well. Finally, the decane/ethanol mixture was decanted and decane/ethanol-filled gels were soaked in 5 mL of water to induce liquid–liquid phase separation of a decane-rich minority phase and an aqueous-rich majority phase within the gel. This last step is performed on a confocal laser scanning microscope for visualization (details in "Optical microscopy"). In dissolution experiments, we first induce phase separation as described above. Then, we decant the water and soak the gels in 5 mL of ethanol while simultaneously performing optical microscopy.

### Fluorescent labeling of agarose
Agarose was fluorescently labeled with 5-([4,6-Dichlorotriazin-2-yl]amino)fluorescein hydrochloride (DTAF), a fluorescein derivative. First, 3 g of agarose was dissolved into 150 mL of H$_2$O. Next, a separately prepared mixture of 30 mg DTAF + 500 mg Na$_2$SO$_4$ + 20 mL H$_2$O was added to the agarose solution. Subsequently, 120 $\mu$L of 10% (w/w) NaOH in water was added to the solution. The mixture was allowed to stir for 2 h at 80 °C, after which an excess of ethanol was added to precipitate the DTAF-labeled agarose. The precipitate was alternately washed with ethanol and water five times before being vacuum dried, crushed, and stored in a vial. All gels were made with 20% wt. DTAF-labeled agarose and 80% wt. unmodified agarose.

### Fluorescent labeling of decane
A fluorescent dye was co-dissolved at a concentration of 0.1 mg/mL in the decane/ethanol solutions during gel soaking. In time-resolved microscopy experiments requiring continuous laser illumination, the bright and photostable dye BDP 558/568 NHS ester from Lumiprobe Corporation was employed. The NHS ester group was unused, and this dye was not bound to any other chemical species. During solvent-exchange condensation, the dye partitioned into the decane-rich phase as solvent exchange proceeded. In experiments with an excess of TX-100 surfactant, BDP 558 was removed from the oil phase via surfactant-induced solubilization. To overcome this limitation, all surfactant experiments were performed with a more hydrophobic dye, diphenylhexatriene (DPH).

## Optical microscopy

Optical microscopy was performed on an inverted confocal laser scanning microscope (Nikon A1R). All imaging was performed using a 60×, 1.4 NA oil-immersion lens. Fluorescence excitation was provided by diode lasers, and signal was detected with GaAsP photomultiplier tube detectors. When performing fluorescent confocal imaging, DTAF-labeled agarose was excited at 488 nm and fluorescence emission was detected with a 500–550 nm bandpass filter. When performing time-resolved measurements with BDP 558 as the oil dye, a 561 nm excitation laser was employed, with a detection window between 570–620 nm. When using DPH as the oil dye, a 405 nm excitation was employed, with a detection window between 425–475 nm. All multi-spectral images were acquired with sequential laser illumination to avoid fluorescence channel bleed-through. Bright-field imaging was performed with a transmitted-light photomultiplier tube detector, with signal acquisition performed during the 561 nm laser excitation period.

## Microscopy image analysis

Microscopy images were denoised using the built-in Nikon Denoise.AI function which is optimized to denoise photomultiplier tube data. All subsequent image analysis was performed using MATLAB® and Fiji/ImageJ (NIH).

To determine the lateral area of condensates in the bright-field microscopy images of Figs. 1 and 2 and Supplementary Fig. 9, a standard deviation filter was used to locate the regions of the image that were in focus (high spatial intensity variation). The filtered images were binarized to obtain the lateral area of the condensates.

In Fig. 3, the condensate image (red channel) was binarized to determine the droplet edge profile (example profile outlined in Fig. 3a). Droplet edge profiles between $t = 0$ and 692 s were plotted in Fig. 3d and were colored according to the local network intensity (green channel). The color scale was normalized to the background network intensity of the undeformed network.

The histograms of condensate circularity in Fig. 6 and Supplementary Fig. 21 were obtained by first binarizing and segmenting condensates from the confocal micrographs in Supplementary Fig. 20. Only condensates above a threshold size of 4 μm$^2$ were included in the analysis. Condensate circularity was measured using the MATLAB "circularity" subroutine in the "regionprops" function, which computes the circularity, $C$, of segmented objects as $C = (4 \cdot \text{Area} \cdot \pi)/(\text{Perimeter})^2$.

## Pendant droplet tensiometry

Pendant droplet tensiometry was performed on a Krüss Drop Shape Analyzer, DSA30. To measure the interfacial tension between decane and water with varying amounts of TX-100 surfactant, a syringe capped with a 20-gauge needle was first loaded with an aqueous solution of TX-100 at the desired concentration. The needle was lowered into a cuvette filled with 2 mL of decane. An aqueous droplet with volume at least 5 μL was then ejected to form a pendant droplet within the oil phase. The droplet shape was allowed to stabilize for at least 2 min before the droplet shape was fitted with the Young–Laplace equation to determine the oil–water interfacial tension, $\gamma_{ow}$.

## Underwater oil contact angle

Underwater oil contact angle measurements were performed on a Krüss DSA30. To measure the underwater contact angle of decane on agarose, an agarose slab cast from a 2.0% w/w solution was floated on top of water. Underneath this slab, a hooked needle was used to inject a 31 μL droplet of decane which floated up to contact the agarose slab. The droplet shape was fitted in the captive bubble orientation with the ellipse (tangent-1) fitting method.

## Cross-sectional scanning electron microscopy

Cross-sectional scanning electron microscopy (SEM) was performed on a Verios XHR SEM. Samples were prepared for cross-sectional SEM imaging by first imbibing and condensing oil droplets in gels of varying concentration. These gels were then plunge frozen in liquid nitrogen-cooled liquid ethane and subsequently both the water and decane phases were removed with a Labconco FreeZone Triad shelf lyophilizer. Lyophilized samples were fractured to expose their cross sections. Samples were mounted to SEM stubs for imaging and a 3 nm iridium coating was deposited with a Leica EM ACE600 magnetron sputter coater. SEM imaging was performed at 2 kV.

## Data availability

The data that support the findings of this study are available within the article and its Supplementary Information. Additional relevant information is available from the corresponding author upon request.

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

## Acknowledgements

The authors thank R. K. Prud'homme, C. P. Brangwynne, H. A. Stone, N. Bizmark, D. L. Chase, D. M. Scott, N. Caggiano, K. Randazzo, and J. Schneider for their insightful discussions. This work was supported by the National Science Foundation (NSF) Materials Research Science and Engineering Center Program through the Princeton Center for Complex Materials (PCCM) (DMR-2011750) (R.D.P.). The authors acknowledge the use of Princeton's Imaging and Analysis Center, which is partially supported through the Princeton Center for Complex Materials (PCCM), a National Science Foundation (NSF)-MRSEC program (DMR-2011750) (R.D.P.).

## Author contributions

J.X.L., M.P.H., C.B.A, and R.D.P. formulated the idea and scope of work. J.X.L. executed the experiments. J.X.L. wrote the manuscript with support from M.P.H., C.B.A., A.K., S.S.D, and R.D.P. All authors contributed to analysis, discussion, and preparation of the manuscript.

## Competing interests

The authors declare no competing interests.
