## [Peer Review File · Nature Communications]

REVIEWER COMMENTS

Reviewer #1 (Remarks to the Author):

In this work, the authors investigate how true liquid-liquid phase separation, influenced by solvent exchange and the growth of an oil-rich phase is influenced by the fibrillar network within which the droplet is embedded. They uncover a series of interesting findings. The condensates being studied are of roughly the same size as the meshwork created by the fibrillar networks. The authors find that the shapes of condensates, the stresses that propagate through the fibrillar networks, the buckling and bending of the network, and the structure of the network is impacted by the interfacial tension (modulated by the addition of a surfactant) and the network structure / rigidity, which is modulated by changing the extent of networking realized by changing the concentration vis-a-vis the gel point. Overall, the findings are extremely interesting and of considerable relevance to a diverse group of investigators in biology, living matter, and soft matter. Tellingly, I have very minor comments / suggestions and no major revisions to request. Here are the points that came up during my reading of the MS:

1) In general, when connecting to the topic of biomolecular condensates, it might be useful to take a more nuanced view. At this juncture, there is growing awareness that a) condensates form via processes that go beyond simple LLPS (please see: <https://doi.org/10.1021/acs.chemrev.2c00814> and <https://doi.org/10.1016/j.molcel.2022.05.018>). Further, condensates are most certainly viscoelastic rather than purely Newtonian materials. So, accommodations for these nuances in the introduction and discussion section will be useful.

2) This sentence, especially the latter half took a while to parse: "However, while these systems exhibit a rich phenomenology and potential for application, only continuum-level mechanics are permitted since condensates necessarily probe bulk network properties in these gels: " Please expand on precisely what is intended. I think I understand but spelling it out would be helpful for the average reader (I count myself in this group).

3) The bright field images in Figure 1 are not easy to parse or process visually. My biased reading was that the images in Figure 2 were more helpful, as were the movies. Therefore, my recommendation is that these images be used as supplemental materials rather than the main drivers of the quantitative analyses.

4) The interface jumps remind one of adsorption transitions that one would liken to prewetting. Is there a parallel that is being missed? In other words, is there a contribution from such an effect, especially in the early stages, or is this truly a non-wetting system on all length and timescales? If so, then it appears that there is an analog of the adsorption transitions, perhaps referable to as desorption transitions, that are in play here that one could think of as pre-dewetting.

5) It would help to define the term mesh-scale condensates before using it.

6) This caption for Figure 2 was confusing: "Confocal fluorescence microscopy time series of an oil condensate, (a) in red, as it fractures the agarose network, (b) in green".

7) I think the content of Fig. 2e needs a deeper discussion to help the reader. What is currently written "As the cavity expands beyond the mesh size, the network is compacted into a shell whose density, and thus fluorescent brightness, increases with deformation extent (Fig. 2e)." does not help the reader much and there are deeper insights to be extracted that are alluded to later in the text.

8) Finally, it took me a while, and I think I am still just guessing, to determine which histogram went with which image in Figure 4. Please help the reader both in terms of the layout of the figure and with the figure caption.

Reviewer #2 (Remarks to the Author):

The authors use experiments to study liquid-liquid phase separation (LLPS) within a fibrillar network. This is a topic of substantial current interest in biology in soft matter. The experiments and the analysis appear to be careful and thorough, the results are novel and interesting, the figures are helpful, and the writing is relatively clear -- however, I think numerous aspects need to be clarified and/or elaborated on, as detailed below. The manuscript should certainly be reconsidered after revision to address these comments.

- In the introduction, it would be helpful for the authors to introduce the basic phenomenology of this problem: condensates grow until they reach the mesh size, at which point they can only continue growing if they either invade the mesh (unfavorable due to capillarity because the condensate is nonwetting to the mesh) or distort the mesh (unfavorable due to mechanical resistance). Providing this description would also give the authors an opportunity to introduce and define capillarity as relevant to this system (it is currently assumed that the reader will already know what "condensate capillarity" means), to introduce the mechanics of fibrillar networks, and to more carefully elaborate on the differences between fibrillar networks and rubbery gels (see next point). This discussion can link directly to Fig 1a, which nicely illustrates the relevant processes but isn't fully explained in the text.

- In the introduction and throughout the manuscript, the authors frame LLPS in fibrillar networks as being fundamentally different from LLPS in "rubbery polymer gels". Yet, both materials are gels comprising a network of polymers saturated with water, so even this basic terminology is confusing and ambiguous. I will refer to these two materials as "fibrillar networks" and "rubbery gels". Ultimately, these two systems seem to differ in two important ways:

(1) fibrillar networks (eg, agarose) are not covalently crosslinked, unlike rubbery gels (eg, silicone gels). Hence, these two kinds of gels exhibit quite different mechanical behavior. Rubbery gels are highly elastic, whereas (as we eventually come to learn) fibrillar networks deform readily and then mostly plastically.

(2) fibrillar networks have a much larger mesh size (a few hundred nm to a few μm), which has two important consequences. First, sub-mesh-size condensates are easily visible. Second, capillarity is comparable to the strength of the network, so the condensate can actually invade the mesh. In rubbery gels, the mesh size is typically much smaller and thus capillarity is much stronger, so mesh invasion is unlikely (and, in any case, would be extremely difficult to image due to the small mesh size).

All of the findings in this study stem from these two features of fibrillar networks. These features themselves are not findings of the study -- they are known from the outset. The results are still novel and fascinating, but these foundational points should be explained clearly and explicitly at the beginning. The last paragraph of the introduction attempts to do this, but does not achieve it. The sentence on lines 57-58 says that fibrillar networks are "gels where condensates and fibrillar strands are commensurate in size", whereas the sentence on line 65 refers to "the commensurate size of oil condensates and the pore space"; these two statements are not the same and I think the latter is more accurate/relevant, but neither actually addresses the key point (see 2 above). The authors should also explain why/how these key differences are thought to be relevant to cells.

- The last paragraph of the introduction also mentions very briefly the method of triggering phase separation: "Experimentally, we imbibe agarose hydrogels with dilute mixtures of n-decane in ethanol (Fig. 1a,i). Subsequently, we induce phase separation via solvent exchange, in which we diffusively displace the imbibed ethanol with water, a poor solvent for decane (Fig. 1a,ii)." I can only guess at what is actually happening here. Some questions:

1. What does it mean to "imbibe" an agarose hydrogel with ethanol? Imbibition is usually a spontaneous process in which capillarity draws a preferentially wetting fluid into a porous material. Please explain.
2. What does it mean to "diffusively displace" the ethanol? In general, I would avoid using the word "displace" to refer to diffusion, which is more of a spreading/exchange process. Is it simply that the gel is immersed in water?
3. Why doesn't the decane get "displaced" along with the ethanol?
4. On what timescale does this displacement process occur relative to the rate of phase separation?
5. What controls the rate of phase separation and the total volume of condensates that form?
6. What sets the final size of the condensates?

7. The supplementary information notes that the exchange of water for ethanol does not change the degree of swelling. Is the same also true for decane, or can the formation of decane condensates cause local changes in the network structure via osmotic effects?

The authors refer variously to "biomolecular condensates", "membraneless organelles", "liquid protein condensates", and "droplets". None of these terms are explicitly introduced or explained. In addition, some consistency would be appreciated -- they are all being used to refer to essentially the same thing.

lines 34-35: "in bulk viscous media" -- A little vague and awkward. Is this a reference to liquid-liquid phase separation in the presence of a third liquid, or just to the separation of two liquids, or something else?

lines 37-38: "within complex media" -- Again, a little vague and awkward. I assume that the authors are referring to liquid-liquid phase separation in the presence of a dispersed solid phase, such as within a porous network?

lines 52-53: "only continuum-level mechanics are permitted since..." -- This sentence is confusingly structured. There is a "since" phrase and then an independent clause after a colon, both of which seem to be explaining the introductory phrase. I suggest moving the part after the colon to the beginning and then dropping the bit before the "since".

line 63: "and we find" -- As far as I can tell, the two halves of this sentence are unrelated. I suggest replacing "and" with a period.

Figure 1:

- It would be nice if there were a clearer link between the 6 panels in Fig 1a and the six panels in Fig 1b. They do not correspond.

- In the caption, the terms "lateral area" and "vertical width" are confusing since "lateral" and "vertical" are often orthogonal directions, and neither direction is actually defined anyway. I think "lateral area" can probably be called "in-plane area" instead. Additionally, the orange markers are faint and pixel-y.

line 82: "permeates the network pore space (Fig. 1b ...)"

- The condensate size is clear from the images, but what is the mesh size in this example?

- The image quality in Fig 1b is not great. It is very difficult to see whether the condensates are indeed permeating the network, as the authors claim, or if cavities have already been formed at this point. The

fluorescence imaging in Fig 2 could presumably be used to give a clearer indication. The idea that the non-wetting condensate can permeate the mesh and grow via Haines jumps is a key argument at the heart of this paper, and so it should be absolutely clear that this is indeed what is happening.

lines 87-89: "These rates are substantially greater than the overall fluid condensation rate of 0.3 fL/s (Supplementary Text 2), implying that the jumps are capillarity driven." -- This needs some explanation. Why might one expect capillarity to drive sudden bursts? (I know the answer, but it should be explained here.)

line 94: "on the liquid" -- Which liquid?

line 95: "the confining cage of agarose fibrils expands, constraining the condensate" -- Cause and effect are backward here; the expansion of the cage does not constrain the condensate.

lines 96-101: This is a description of classical, quasi-static drainage leading to a Haines jump. Some references would be appropriate (probably not reference 30?).

lines 102-106: It is a shame that these processes are not illustrated in the paper, but I appreciate that they are illustrated in the supplemental material.

line 108: "deformable matrix^{31,32}" -- Should be rigid matrix? The systems in Refs 31 and 32 are not deformable.

lines 108-109: "where the pervasion... solvent exchange" -- See above about explaining in more detail how this works.

line 112: Does Ref 35 belong here?

line 114: "at $Ca \ll 1^{34}$." -- I suggest writing as " $(Ca \ll 1)^{34}$ " to avoid the reference looking like an exponent.

lines 119-121: "Due to the subisostatic architecture of agarose networks, the initial mechanical response to capillary forces primarily arises from the bending rigidity of adjacent fibrils and is comparatively soft,

dictated by the shear modulus G ." -- Soft compared to what? Can you give a number? Is it a bending stiffness or a shear modulus? If the latter, shear modulus of what? Also, please briefly define/explain "subisostatic".

line 124: How does the stiffness change (quantitatively) between the bending and stretching regimes?

lines 131-135: Here and throughout, I think that more clarity is needed with regard to the mechanics of the fibrils and their different deformation mechanisms (elastic bending, elastic stretching, plastic yielding (relative sliding?), and fracture).

- It feels like "deform", "fracture", and "yield" are often used interchangeably, but they are distinct processes.

- I think there is a small amount of elastic bending, followed by plastic yielding and then fracture with a negligible amount of stretching; is that right? Please explain in more detail.

- A yield stress of ~ 340 MPa is quoted, which seems large. The term "yield" usually refers to plasticity, which I think is happening much of the time here. This value seems like more of a tensile strength? Please clarify and avoid using "yield" when referring to fracture.

- I'm not too sure about the calculation of this stress (Supplementary Text 7). I think a classic thin-walled pressure-vessel argument would be cleaner $[(2R)(2\gamma\cos) \sim 2\pi r_f^2 \sigma]$. Note that the result is a factor of 2π smaller.

- line 136: "which fracture at strains below 50%" -- What does the strain have to do with it? The value quoted in the previous sentence was a stress.

lines 139-141: "This process is mechanistically different in fibrillar gels since topologically, the fracture strength of individual network elements must be exceeded to permit cavity formation." -- Why is this not also true of rubbery gels?

line 147: "dissipation energy density" -- First mention of this concept. What is this?

lines 147-149: This sentence could be worded more clearly.

Figure 2:

- Panel d: Consider making the trajectories white.

- Caption: The first sentence is difficult to parse. Consider splitting the descriptions of a and b into separate sentences.

- Fig 2e: The symbol rho for 'Network spatial intensity profile' is not mentioned anywhere in the caption or in the text.

Section "Cavity growth and network deformation" -- This section finally provides a lot of helpful information on the mechanics of the fibrillar network. Why does it come *after* the section on network fracture?

line 169: "material density"  "network density" ?

line 170: "network resistance"  "mechanical resistance from the network" ?

line 170: "Upon fracture, locally weaker regions of the network yield first" -- Not necessarily? Some stronger regions might experience larger stresses.

lines 182-184: "while the black trajectory..." -- I found this bit difficult to parse in connection with the image.

line 186: "of molecular gels, whose networks do not accumulate substantially around growing droplets" -- I was surprised by this. Can the authors explain why this might be? Separately, I assume that "molecular gels" are rubbery gels?

line 187: This value of nu is basically zero, which seems plausible. Some error bars would be appreciated. I'm concerned that the associated calculation (Supplementary Text 8) will be extremely insensitive to nu for such small values, for which beta is very close 1. Separately, something is missing from the explanation -- one cannot get from E_{el} and W to the ODE for s without at least one more equation.

lines 192-194: "Cross-sectional scanning..." -- I think this measurement is done after removing the droplet? The caption of Fig 3 refers to liquid ethane plunge freezing and lyophilization, but I'm not sure what those processes are or what they do. Is there any relationship to the re-dissolution of the droplet described on lines 203-204? Please elaborate.

line 196: "non-affine" -- I'm not sure that this is the right term here. "Localised" might be better.

line 197: "a barrier"

- This barrier looks like the bridge that fails in 2b1-b3.

- Would it be possible for the decane to re-dissolve into the surrounding fluid and diffuse through this barrier, as during Ostwald ripening?

lines 199-203: "Semiflexible polymer fibrils such as agarose deform plastically..."

- Why are we only now learning this??? This belongs in the introduction.

- Again, it is unclear how the different mechanical processes match up. Here, the authors note that agarose deforms plastically, but then say that this results in bending rigidity dominating. The discussion in lines 120-136 suggests that plastic deformations occur independently of elastic deformations related to the bending rigidity?

- Which is more relevant to biological cells, the reversible deformation of rubbery gels or the permanent plastic deformation of fibrillar networks?

line 208: "excess surfactant" -- In what sense "excess"? I think this only becomes clear later, when surfactant is discussed.

Fig 3d/e: Please check the caption. The decane condensate is red in 2e, not 2d.

226-227: How does the presence of surfactant "preclude network fracture". Is it just by reducing the strength of capillarity, and hence reducing the incentive for the non-wetting decane to separate from the network? This sentence reads as if the surfactant has some direct chemical influence on the network fibrils themselves, such as increasing their fracture toughness.

228: How do you know whether or not there has been network fracture? Is this just inferred from the shape of the final condensate, or is it visible in the fluorescence imaging?

lines 230-231: "with no added surfactant, interfacial tension dominates and spherical droplets are exclusively observed (Fig. 4e)." -- In that case, did all of the preceding observations involve some surfactant? Please clarify!

lines 239-240: "solute is already fully depleted prior to network fracture" -- This is the first reference to solute depletion. Please elaborate, ideally in the introduction during a general overview of the process (see above).

Fig 4a: This figure demonstrates some interesting behaviour, but suffers from being rather small - it is hard to make out the different morphologies clearly.

Reviewer #3 (Remarks to the Author):

It was a real pleasure to read this manuscript. The experiments are very creative. The phenomena are rather complex, but the authors have done an excellent job of reducing their observations to their key elements and representing the data visually (Fig 2d is gorgeous and clear). The text is excellent – simultaneously compact and lucid.

The problem is fascinating. The competition of condensation, capillarity, and elasticity is unexplored, especially in heterogeneous systems, and the authors make a compelling connection both to recent questions in biophysics as well as more classical phenomena in rigid porous media.

I support publication in Nature Communications provided that the authors can address the following points:

1. I am not sure what terminology should be employed to correctly describe their elastic matrix. The authors describe it as fibrillar, but that doesn't seem accurate. To me a fibrillar network would have clearly resolvable linear structures with a persistence length larger than the mesh scale. (like a collagen gel). Here, it looks more like a very heterogeneous molecular gel. Like something that cross-linked while phase separating. In no way does this detract from the significance of the manuscript. The essential point of the manuscript is that they are probing condensation in the limit where the droplets are at a scale that samples network heterogeneity. I think the authors should no longer refer to this as a fibrillar network, and adjust the terminology throughout the rest of the paper accordingly. (NOTE: my understanding is that lyophilized gels tend to look pretty fibrillar in EM even if the swollen networks are not) If the authors insist on fibrillar they will need better in situ structural characterization – maybe with super resolution optical imaging???

2. Lines 119-121. In the same sentence, the authors refer to the structure being sub-isostatic but having a finite shear modulus. This seems contradictory. Please clarify.

3. Because of my comments in point 1, I think that the discussion of the yielding of 'fibrils' and its comparison to collagen (lines 130ish) is dodgy and should probably be removed.

Reviewer #4 (Remarks to the Author):

Review of Liu et. al. "Liquid-liquid phase separation within fibrillar networks"

The study aims to understand the effects of fibrillar networks on liquid-liquid phase separation occurring in them. The condensates formed inside networks with commensurate mesh-size grow sporadically in bursts. The deformation of the network caused by the formation and growth of condensates is permanent. The dynamics of condensate growth is dependent on the condensate capillarity and network strength. The material of the paper is relevant to the field and is clearly written.

But there are some significant concerns regarding the scope, analyses and presentation. The primary novelty of the paper is in studying the sporadic growth dynamics of the condensates in the network. Multiple publications previously have studied condensates inside networks; in fact quite a few of such publications have been cited here. For example, in ref 13 Fig. 4b, and ref 16 shows droplets with similar size and the mesh in actual biological contexts. Ref 22 discusses the physics of droplet formation inside networks and the forces they exert on the fibers. Coupled with these existing concepts, the sporadic growth has the potential to shed light on key aspects of phase separation inside networks.

Unfortunately, the manuscript has been presented entirely based on observations without contextualizing them or answering any questions that have been raised in the introduction section as the motivation of the study which fails to communicate the significance of the results. For the above reasons I do not believe this manuscript is ready for publication.

Specific concerns:

Introduction:

Line 45: Reference 21 is a review paper. The authors should mention specific problems in probing droplets in vivo that their experimental model can address. This is especially necessary since there exists a large body of literature on probing droplets in vitro.

Line 52: The wording "continuum-level mechanics are permitted since condensates and molecular strands" is slightly confusing. Maybe the language can be clarified.

Line 68: Please provide a definition/ description for condensate capillarity.

Results:

Overall: Most of the calculations done are based on a single dataset which can be a source for unintended bias. A more statistically robust approach is required.

Figure 1: Why are all the lobes smaller than the mesh size calculated in Supplementary Text 1? Given the standard deviation shown in Fig. S3, some lobes should be substantially larger. Also, time points should be mentioned in Fig. S4 to avoid confusion. Are the 4 points in Fig. S4A same as Fig.1 b1, b2, b3 and b6? If so, then why does the placement of the red dots corresponding to b2 and b3 seem different in Fig. S4?

Line 83: Tracking just the lateral area and interpolating the volume can lead to errors. Needs to be addressed. Also, any drift in Z-direction (a common problem with confocal microscopes) can introduce substantial errors in the calculations described in Supplementary Text 2. Given that a confocal microscope is used, Z-stacks should be recorded and the volumes calculated from there.

Line 85: Lobe 1 actually appears around 72s in Supplementary Video 1.

Line 87: Assuming the flowrates are calculated based on the jumps in Fig. 1c, the second jump clearly is not nearly twice times the size of jump 1. The discrepancy in scaling needs to be addressed.

Line 88: How is the condensation rate of 0.3 fL/s calculated? It is not mentioned in Supplementary Text2.

Line 88: What is a capillarity-driven jump? The term needs to be explained adequately before concluding.

Line 96: Mentions Supplementary Text 3. The jump in Fig. S5 happens over 40 seconds as opposed to 2. In Supplementary Video S5, the lobe formation happens between 108 and 110 seconds. What is the rationale for choosing 90 and 130s?

Line 99: If lobe formation is redistribution, then the overall volume of the condensate should not change. This needs to be shown by comparing actual volumes from z-stacks. Corresponding decrease in thickness of another lobe is not conclusive. Another possibility is that formation of a new lobe deforms the network and generates a strain which causes the shrinking of other lobes.

Line 101: Define/ describe what is an interface jump.

Figure 2: Mention agarose concentration. Fig. 1 is 0.3% and Fig. 3 is a mix of 2%, 0.8% and 0.3%. Also, why is the process shown Fig.2 much slower than in Fig. 1? Between a4 and a5, a lot of structures around the bigger condensate seems to disappear. Are they coalescing? That would be antithetical to the argument made in Lines 197-198.

Line 123: "Beyond a critical strain..." This critical strain seems to be the most important factor in condensate growth. This strain should correspond to a particular size ratio of the condensate and mesh-size. The strain should be estimated and rearrangement of the network should be studied.

Line 143: "Subsequent network deformation proceeds via further fracture" is problematic. Why does it not form more lobes? What determines the relative competition between squeezing into a new lobe vs fracturing fibrils to form spherical droplets? For example, in Fig.1 (and Fig. S4) the system keeps adding lobes to the structure whereas in Fig.2 (and Fig. S8) a two-lobe structure fractures the fibrils. This needs to be addressed.

Lines 144-147: The logic here seems inverse. The cavity expansion should be easier (faster) if fracturing strands are easier than dissipating the tension through the network. So fracturing fibrils should become easier with increasing cavity expansion velocity. Also, Fig. S9 does not show increase in velocity in all

cases. It shows a drop from 0.3% to 0.8% which is not even discussed. Again, any analysis based on one dataset is not rigorous.

Lines 168-170: The statement indicates that the expansion of the cavity can be used to probe the microstructure of the network. But no such information is extracted in this work.

Figure S8: Red and green panels are a complete mismatch in C and D. The cavity growth in both these cases is uniform and lacks any tortuosity. Also, cavity expansion velocity is calculated from the condensate fluorescence which complicates the problem further.

Figure 3: Why is the SEM image from a different cgel from the rest of the main manuscript? Fig. S18 seems to have all concentrations. But the thick shell of fiber seen in 2% (and 1.3%) is lacking in 0.3%. This needs to be addressed. More importantly, the mesh-sizes from B show that 0.8% has a larger mesh-size than 0.3%. Might be the issue in just one image, but there are no other images to compare with. Why does panel d lack the fluorescence from the decane (as in panel e)? All conditions should be reported uniformly.

Lines 230-231: Authors claim that for 0.3% agarose gel and no surfactant case all droplets are spherical. Does that mean droplets showed in Fig.1 will eventually become spherical? If so, what is the timescale? Even in a single condition there seems to be a competition as mentioned in the point for Line 143.

Discussion:

This is the section that is most concerning as the authors do not 'discuss' their results. There is no discussion as to this sporadic growth model is in agreement or contradiction with other works. If the network fracture eventually creates spherical droplets larger than the mesh size, then what is the significance of the lobe formation? The authors need to discuss specific questions that can be answered from their observations. It is intuitive that condensates inside a network have to nucleate in a size smaller than the mesh-size and grow by fracturing/deforming the network (as discussed in ref. 22). The authors need to justify the novelty of their results, which I assume is the lobe formation. From Fig. 1a(fourth image) it seems that after a certain amount of interface jumps, the network is embedded inside a tortuous condensate which looks strikingly similar to ref 28, which should be brought into the discussion.

Response to Reviewers:
“Liquid-liquid phase separation within fibrillar networks”
by Liu et al. for Nat. Comms. NCOMMS-23-06794

Below, reviewer comments are in blue and our responses are in black.

Reviewer 1

“In this work, the authors investigate how true liquid-liquid phase separation, influenced by solvent exchange and the growth of an oil-rich phase is influenced by the fibrillar network within which the droplet is embedded. They uncover a series of interesting findings. The condensates being studied are of roughly the same size as the meshwork created by the fibrillar networks. The authors find that the shapes of condensates, the stresses that propagate through the fibrillar networks, the buckling and bending of the network, and the structure of the network is impacted by the interfacial tension (modulated by the addition of a surfactant) and the network structure / rigidity, which is modulated by changing the extent of networking realized by changing the concentration / the gel point. Overall, the findings are extremely interesting and of considerable relevance to a diverse group of investigators in biology, living matter, and soft matter. Tellingly, I have very minor comments / suggestions and no major revisions to request.”

We thank Reviewer 1 for their review of the manuscript and for their helpful comments. We appreciate the Reviewer’s favorable outlook of the manuscript, and in response to the Reviewer’s comments, we have provided additional information in the main text, detailed below. We also address the Reviewer’s specific questions.

Here are the points that came up during my reading of the MS:

1) In general, when connecting to the topic of biomolecular condensates, it might be useful to take a more nuanced view. At this juncture, there is growing awareness that a) condensates form via processes that go beyond simple LLPS (please see: <https://doi.org/10.1021/acs.chemrev.2c00814> and <https://doi.org/10.1016/j.molcel.2022.05.018>). Further, condensates are most certainly viscoelastic rather than purely Newtonian materials. So, accommodations for these nuances in the introduction and discussion section will be useful.

We thank the Reviewer for bringing this up. We have substantially expanded the Discussion section to address these points. Please see the second paragraph of the revised Discussion section (revised lines 407-420) for the specific changes regarding these points.

2) This sentence, especially the latter half took a while to parse: "However, while these systems exhibit a rich phenomenology and potential for application, only continuum-level mechanics are permitted since condensates necessarily probe bulk network properties in these gels: " Please expand on precisely what is intended. I think I understand but spelling it out would be helpful for the average reader (I count myself in this group).

We have made several changes to the Introduction section (revised lines 49-52) to improve clarity.

3) The bright field images in Figure 1 are not easy to parse or process visually. My biased reading was that the images in Figure 2 were more helpful, as were the movies. Therefore, my recommendation is that these images be used as supplemental materials rather than the main drivers of the quantitative analyses.

We thank the Reviewer for this comment. In response to this comment as well as similar ones from the other reviewers, we have made substantial changes to the main text. Fig. 1 now shows fluorescence data and also demonstrates the entire process, from phase separation to network fracture and cavity growth. We have also added a section which describes the entire process (revised lines 100-125).

4) The interface jumps remind one of adsorption transitions that one would liken to prewetting. Is there a parallel that is being missed? In other words, is there a contribution from such an effect, especially in the early stages, or is this truly a non-wetting system on all length and timescales? If so, then it appears that there is an analog of the adsorption transitions, perhaps referable to as desorption transitions, that are in play here that one could think of as pre-dewetting.

This experimental system consists of decane, water, and ethanol. Upon phase separation, the decane-rich phase is quite hydrophobic and is a non-wetting fluid for the agarose network at all times (see Supplementary Note 3). Agarose is a polysaccharide-based biopolymer which is highly hydrophilic. We believe that the interface jumps which we observe are analogous to Haines jumps in porous media, where a non-wetting fluid (the growing oil condensate) invades a porous space (the mesh space of the agarose network) which is wetted by a wetting fluid (water). We believe that when the decane-rich phase first forms, it forms within a void formed by the porous network mesh before growing and becoming constrained by the network. Therefore, we do not believe the condensation or interface jumps arise from any prewetting or pre-dewetting transitions.

5) It would help to define the term mesh-scale condensates before using it.

Thank you, we have clarified the text by re-phrasing this term.

6) This caption for Figure 2 was confusing: "Confocal fluorescence microscopy time series of an oil condensate, (a) in red, as it fractures the agarose network, (b) in green".

We thank the Reviewer for pointing this out; we have revised the caption to this subfigure (revised lines 226). Please note that the original Fig. 2 is now Fig. 3 in the revised text.

7) I think the content of Fig. 2e needs a deeper discussion to help the reader. What is currently written "As the cavity expands beyond the mesh size, the network is compacted into a shell whose density, and thus fluorescent brightness, increases with deformation extent (Fig. 2e)." does not help the reader much and there are deeper insights to be extracted that are alluded to later in the text.

Thank you for pointing this out, we have now expanded the discussion on Fig. 3e (revised lines 262-269) (note that this was previously Fig. 2e).

8) Finally, it took me a while, and I think I am still just guessing, to determine which histogram went with which image in Figure 4. Please help the reader both in terms of the layout of the figure and with the figure caption.

We thank the Reviewer for this suggestion. To help clarify the figure, we have added number markers as well as the schematic condensate morphologies into the blank spaces of the histograms to help identify which histogram goes with which sample. Please note that Fig. 4 is now Fig. 6 in the revised manuscript.

Reviewer 2

The authors use experiments to study liquid-liquid phase separation (LLPS) within a fibrillar network. This is a topic of substantial current interest in biology in soft matter. The experiments and the analysis appear to be careful and thorough, the results are novel and interesting, the figures are helpful, and the writing is relatively clear -- however, I think numerous aspects need to be clarified and/or elaborated on, as detailed below. The manuscript should certainly be reconsidered after revision to address these comments.

We thank Reviewer 2 for their careful review of the manuscript and for their detailed comments. In response to the Reviewer's comments, we have substantially revised the manuscript as well as included new figures. Below, we address the Reviewer's specific questions.

- In the introduction, it would be helpful for the authors to introduce the basic phenomenology of this problem: condensates grow until they reach the mesh size, at which point they can only continue growing if they either invade the mesh (unfavorable due to capillarity because the condensate is non-wetting to the mesh) or distort the mesh (unfavorable due to mechanical resistance). Providing this description would also give the authors an opportunity to introduce and define capillarity as relevant to this system (it is currently assumed that the reader will already know what "condensate capillarity" means), to introduce the mechanics of fibrillar networks, and to more carefully elaborate on the differences between fibrillar networks and rubbery gels (see next point). This discussion can link directly to Fig 1a, which nicely illustrates the relevant processes but isn't fully explained in the text.

We thank the Reviewer for this suggestion, and we have revised the Introduction section to include this discussion (revised lines 53-64).

- In the introduction and throughout the manuscript, the authors frame LLPS in fibrillar networks as being fundamentally different from LLPS in "rubbery polymer gels". Yet, both materials are gels comprising a network of polymers saturated with water, so even this basic terminology is confusing and ambiguous. I will refer to these two materials as "fibrillar networks" and "rubbery gels". Ultimately, these two systems seem to differ in two important ways:

(1) fibrillar networks (eg, agarose) are not covalently crosslinked, unlike rubbery gels (eg, silicone gels). Hence, these two kinds of gels exhibit quite different mechanical behavior. Rubbery gels are highly elastic, whereas (as we eventually come to learn) fibrillar networks deform readily and then mostly plastically.

(2) fibrillar networks have a much larger mesh size (a few hundred nm to a few μm), which has two important consequences. First, sub-mesh-size condensates are easily visible. Second, capillarity is comparable to the strength of the network, so the condensate can actually invade the mesh. In rubbery gels, the mesh size is typically much smaller and thus capillarity is much stronger, so mesh invasion is unlikely (and, in any case, would be extremely difficult to image due to the small mesh size).

All of the findings in this study stem from these two features of fibrillar networks. These features themselves are not findings of the study -- they are known from the outset. The results are still novel and fascinating, but these foundational points should be explained clearly and explicitly at the beginning. The last paragraph of the introduction attempts to do this, but does not achieve it. The sentence on lines 57-58 says that fibrillar networks are "gels where condensates and fibrillar strands are commensurate in size", whereas the sentence on line 65 refers to "the commensurate size of oil condensates and the pore space"; these two statements are not the same and I think the latter is more accurate/relevant, but neither actually addresses the key point (see 2 above). The authors should also explain why/how these key differences are thought to be relevant to cells.

We thank the Reviewer for raising these detailed points. We have incorporated several parts of this discussion into the Introduction section.

- The last paragraph of the introduction also mentions very briefly the method of triggering phase separation: "Experimentally, we imbibe agarose hydrogels with dilute mixtures of n-decane in ethanol (Fig. 1a,i). Subsequently, we induce phase separation via solvent exchange, in which we diffusively displace the imbibed ethanol with water, a poor solvent for decane (Fig. 1a,ii)." I can only guess at what is actually happening here.

We have added additional text in the Introduction which describes in more detail the experimental protocol and process (revised lines 65-79).

Some questions:

We thank the Reviewer for these questions, and we have revised the text to clarify all of these points.

1. What does it mean to "imbibe" an agarose hydrogel with ethanol? Imbibition is usually a spontaneous process in which capillarity draws a preferentially wetting fluid into a porous material. Please explain.

We used the term "imbibe" to refer to the diffusion of ethanol/decane into the agarose hydrogel, but we realize that we misused the term and have changed the wording.

2. What does it mean to "diffusively displace" the ethanol? In general, I would avoid using the word "displace" to refer to diffusion, which is more of a spreading/exchange process. Is it simply that the gel is immersed in water?

Yes, in the experiments, the gel is submerged under a large volume of the appropriate mixture or liquid, and the solvents exchange by diffusion. We have removed usage of "displace".

3. Why doesn't the decane get "displaced" along with the ethanol?

Decane does not diffuse outwards because of its low solubility in the aqueous phase.

4. On what timescale does this displacement process occur relative to the rate of phase separation?

Solvent exchange from ethanol to water takes place over the course of about 1 hour, whereas the oil solute fully phase separates within 5 minutes in all experiments.

5. What controls the rate of phase separation and the total volume of condensates that form?

The rate of phase separation is determined by the supersaturation and diffusion coefficient of the oil solute. The total volume of condensates is set by the starting concentration of decane in ethanol, which is 4% v/v in all experiments.

6. What sets the final size of the condensates?

The final size of the condensates depends on the nucleation rate of condensates, which is then set by the solvent exchange rate. The greater the solvent exchange rate (e.g. by using a thinner gel such that solvent exchange via diffusion is faster), the greater the nucleation rate, and in that case, while there are a greater number of condensates, they are on average smaller.

7. The supplementary information notes that the exchange of water for ethanol does not change the degree of swelling. Is the same also true for decane, or can the formation of decane condensates cause local changes in the network structure via osmotic effects?

We do not expect decane condensates to locally change the network structure through osmotic effects because decane is a non-wetting phase for the agarose fibrils. We only expect decane condensates to interact with the surrounding fibrillar elements through capillary forces.

The authors refer variously to "biomolecular condensates", "membraneless organelles", "liquid protein condensates", and "droplets". None of these terms are explicitly introduced or explained. In addition, some consistency would be appreciated -- they are all being used to refer to essentially the same thing.

We thank the Reviewer for pointing this out, we have standardized the language to refer only to "biomolecular condensates" and "condensates".

lines 34-35: "in bulk viscous media" -- A little vague and awkward. Is this a reference to liquid-liquid phase separation in the presence of a third liquid, or just to the separation of two liquids, or something else?

Thank you, we have removed this phrase. There is just the phase separation of the decane-rich and the decane-poor liquids.

lines 37-38: "within complex media" -- Again, a little vague and awkward. I assume that the authors are referring to liquid-liquid phase separation in the presence of a dispersed solid phase, such as within a porous network?

We have changed the wording here to be more specific (revised lines 35-36).

lines 52-53: "only continuum-level mechanics are permitted since..." -- This sentence is confusingly structured. There is a "since" phrase and then an independent clause after a colon, both of which seem to be explaining the introductory phrase. I suggest moving the part after the colon to the beginning and then dropping the bit before the "since".

We have changed the wording as well as sentence structure for clarity here (revised lines 49-52).

line 63: "and we find" -- As far as I can tell, the two halves of this sentence are unrelated. I suggest replacing "and" with a period.

Thank you, we have updated the text here (revised lines 70-74).

Figure 1:

Based on the Reviewer's comments, we have made substantial changes to Fig. 1. We have performed additional experiments and we now present a different data set in Fig. 1 which depicts the entire process, from phase separation, to network deformation, to fibril fracture, to network compaction. We have also added a section to the main text which describes this process (revised lines 100-125).

- It would be nice if there were a clearer link between the 6 panels in Fig 1a and the six panels in Fig 1b. They do not correspond.

Now, the new experimental data demonstrates more aspects of Fig. 1a, which we have highlighted using the dashed black lines in Fig. 1.

- In the caption, the terms "lateral area" and "vertical width" are confusing since "lateral" and "vertical" are often orthogonal directions, and neither direction is actually defined anyway. I think "lateral area" can probably be called "in-plane area" instead. Additionally, the orange markers are faint and pixel-y.

We thank the Reviewer for these suggestions, and we have incorporated them into the revised text.

line 82: "permeates the network pore space (Fig. 1b ...)"

- The condensate size is clear from the images, but what is the mesh size in this example?

- The image quality in Fig 1b is not great. It is very difficult to see whether the condensates are indeed permeating the network, as the authors claim, or if cavities have already been formed at this point. The fluorescence imaging in Fig 2 could presumably be used to give a clearer indication. The idea that the non-wetting condensate can permeate the mesh and grow via Haines jumps is a key argument at the heart of this paper, and so it should be absolutely clear that this is indeed what is happening.

As mentioned, based on the Reviewer's comments, we have made substantial changes to Fig. 1. The new data now include fluorescence data for the oil phase and fibrillar network, as suggested by the

Reviewer. In this data, we can clearly see that at early times, the condensate permeates throughout the fibrillar network without fracturing it.

lines 87-89: "These rates are substantially greater than the overall fluid condensation rate of 0.3 fL/s (Supplementary Text 2), implying that the jumps are capillarity driven." -- This needs some explanation. Why might one expect capillarity to drive sudden bursts? (I know the answer, but it should be explained here.)

We have clarified this in the revised text (revised lines 136-156).

line 94: "on the liquid" -- Which liquid?

This line has been removed in the revised text.

line 95: "the confining cage of agarose fibrils expands, constraining the condensate" -- Cause and effect are backward here; the expansion of the cage does not constrain the condensate.

Thank you, we have fixed this.

lines 96-101: This is a description of classical, quasi-static drainage leading to a Haines jump. Some references would be appropriate (probably not reference 30?).

We have now updated the references.

lines 102-106: It is a shame that these processes are not illustrated in the paper, but I appreciate that they are illustrated in the supplemental material.

Based on the Reviewer's comments, we have included a new Fig. 2 in the revised manuscript which demonstrates these processes.

line 108: "deformable matrix^{31,32}" -- Should be rigid matrix? The systems in Refs 31 and 32 are not deformable.

We thank the Reviewer for pointing this out. We have updated the wording to reflect this (revised lines 157-163).

lines 108-109: "where the pervasion... solvent exchange" -- See above about explaining in more detail how this works.

Thank you, we have elaborated on this in the new section added to the revised text (revised lines 100-125).

line 112: Does Ref 35 belong here?

We have removed this reference.

line 114: "at $Ca \ll 1^{34}$." -- I suggest writing as " $(Ca \ll 1)^{34}$ " to avoid the reference looking like an exponent.

Thank you, we have moved the 34 outside the period.

lines 119-121: "Due to the subisostatic architecture of agarose networks, the initial mechanical response to capillary forces primarily arises from the bending rigidity of adjacent fibrils and is comparatively soft, dictated by the shear modulus G ." -- Soft compared to what? Can you give a

number? Is it a bending stiffness or a shear modulus? If the latter, shear modulus of what? Also, please briefly define/explain "subisostatic".

We have substantially changed the language here (revised lines 181-185).

line 124: How does the stiffness change (quantitatively) between the bending and stretching regimes?

Bulk measurements of fibrillar gels show that the stiffness changes by nearly an order of magnitude during the strain-stiffening of fibrillar networks in which fibrils transition from bending to stretching deformations. In agarose specifically, this same extent of stiffening is experienced. We have updated the text and added references to discuss this (revised lines 193-198).

lines 131-135: Here and throughout, I think that more clarity is needed with regard to the mechanics of the fibrils and their different deformation mechanisms (elastic bending, elastic stretching, plastic yielding (relative sliding?), and fracture).

We thank the Reviewer for this comment. We have reworded the section to improve clarity (revised lines 199-211 and others).

- It feels like "deform", "fracture", and "yield" are often used interchangeably, but they are distinct processes. - I think there is a small amount of elastic bending, followed by plastic yielding and then fracture with a negligible amount of stretching; is that right? Please explain in more detail.

Yes, this is correct. We have updated the discussion in the revised text to reflect this (revised lines 179-198).

- A yield stress of ~340 MPa is quoted, which seems large. The term "yield" usually refers to plasticity, which I think is happening much of the time here. This value seems like more of a tensile strength? Please clarify and avoid using "yield" when referring to fracture.

Yes, it is a tensile strength that we refer to, and we have changed the language to reflect these suggestions (revised lines 199-211).

- I'm not too sure about the calculation of this stress (Supplementary Text 7). I think a classic thin-walled pressure-vessel argument would be cleaner $[(2R)^*(2\gamma\cos) \sim 2\pi r_f^2\sigma]$. Note that the result is a factor of 2π smaller.

We believe this geometry, with a fibril forming a hoop around the waist of the condensate, is accurate because of the experimental observations of single fibrils restraining condensates around their centers (e.g. revised Fig. 3b₃ and Fig 1b₅ green channel).

- line 136: "which fracture at strains below 50%" -- What does the strain have to do with it? The value quoted in the previous sentence was a stress.

We have removed the comparison with collagen.

lines 139-141: "This process is mechanistically different in fibrillar gels since topologically, the fracture strength of individual network elements must be exceeded to permit cavity formation." -- Why is this not also true of rubbery gels?

This is indeed also true of rubbery gels. What we meant was that, while in rubbery gels a bulk network property is what dictates cavity growth (a critical driving pressure of $p_c = 5E/6$ where E is the Young's modulus), here, it is the fracture mechanics of individual restraining fibrils which dictates cavity formation. We have modified the wording to reflect this (revised lines 212-223).

line 147: "dissipation energy density" -- First mention of this concept. What is this?

We have removed this.

lines 147-149: This sentence could be worded more clearly.

We have rephrased this (see revised lines 212-223).

Figure 2:

- Panel d: Consider making the trajectories white.

Thank you, we have done so.

- Caption: The first sentence is difficult to parse. Consider splitting the descriptions of a and b into separate sentences.

Thank you, we have reworded the caption (revised lines 226). Note that the previous Fig. 2 is now Fig. 3 in the revised text.

- Fig 2e: The symbol ρ for 'Network spatial intensity profile' is not mentioned anywhere in the caption or in the text.

We have now defined ρ in both the main text as well as caption.

Section "Cavity growth and network deformation" -- This section finally provides a lot of helpful information on the mechanics of the fibrillar network. Why does it come *after* the section on network fracture?

Thank you, we have added discussion of the mechanics of the fibrillar network earlier in the main text (revised lines 179-185).

line 169: "material density"  "network density" ?

We have changed this.

line 170: "network resistance"  "mechanical resistance from the network" ?

We have changed this.

line 170: "Upon fracture, locally weaker regions of the network yield first" -- Not necessarily? Some stronger regions might experience larger stresses.

We have changed the wording just to reflect that anisotropic cavity growth occurs (revised lines 253-255).

lines 182-184: "while the black trajectory..." -- I found this bit difficult to parse in connection with the image.

We have clarified the text (revised lines 269-279).

line 186: "of molecular gels, whose networks do not accumulate substantially around growing droplets" -- I was surprised by this. Can the authors explain why this might be? Separately, I assume that "molecular gels" are rubbery gels?

Due to the entropic mechanical response of rubbery gels (molecular gels), their deformations are largely affine, and rubbery gels have a Poisson's ratio that is typically near $\nu \approx 0.5$. Therefore, when a cavity grows within a rubbery gel, the network strands affinely deform away from the cavity, preventing substantial network accumulation or densification from occurring around the droplet. This for example is seen in the Fig. S3 of the supporting information of (<https://www.science.org/doi/10.1126/sciadv.aaz0418>) (reproduced here). When we take an intensity profile along the width of this cavity using ImageJ (for example along the horizontal yellow line), we see that the slightly brighter shell around these droplets only has an intensity 1.2x the background intensity, despite having a network stretch of $\lambda = D/\xi$ over 1000 (where D is the diameter of the droplet ($\sim 10 \mu\text{m}$) and ξ is the mesh size ($\sim 10 \text{nm}$)). Comparatively, in the agarose fibrillar networks here, we show a shell densification of $\sim 2\text{x}$ for just $\lambda \approx 10$.

Uncrosslinked silicone
Fig. S3. CARS microscopy images of fluorinated oil droplets in silicone gels of different stiffnesses. Droplets of fluorinated oil in uncrosslinked silicone oil (left) and silicone gel with $E = 71\text{kPa}$ (right).

line 187: This value of ν is basically zero, which seems plausible. Some error bars would be appreciated. I'm concerned that the associated calculation (Supplementary Text 8) will be extremely insensitive to ν for such small values, for which β is very close 1. Separately, something is missing from the explanation --one cannot get from E_{el} and W to the ODE for s without at least one more equation.

We have caught a mistake we have made, which is that previously our network stretch, λ , was normalized to the mesh size diameter, ξ , rather than the mesh size radius, $\xi/2$. Therefore, you can see that the values of λ against which we plot the new Fig. 3e,f is double that of the previous Fig. 2e,f. With these new values of λ , we faced substantial numerical difficulties with the Neo-Hookean fit, and so we have removed the Neo-Hookean analysis from this manuscript.

lines 192-194: "Cross-sectional scanning..." -- I think this measurement is done after removing the droplet? The caption of Fig 3 refers to liquid ethane plunge freezing and lyophilization, but I'm not sure what those processes are or what they do. Is there any relationship to the re-dissolution of the droplet described on lines 203-204? Please elaborate.

Yes, these SEM measurements are done after removing the droplet. In the text, we have clarified what exactly we have done (revised lines 284-287). No, these experiments do not have any relation to the dissolution experiments.

line 196: "non-affine" -- I'm not sure that this is the right term here. "Localised" might be better.

Thank you, we have changed this.

line 197: "a barrier"

- This barrier looks like the bridge that fails in 2b1-b3.

It is not, because the two droplets are not connected in Fig. 4c. Bridges that fail are always restraining contiguous bodies.

- Would it be possible for the decane to re-dissolve into the surrounding fluid and diffuse through this barrier, as during Ostwald ripening?

Ostwald ripening is certainly possible in this suspension of oil droplets within an aqueous continuous phase. However, we do not observe any Ostwald ripening effects over the course of any of our experiments, which last up to 2 hours. In some cases, we have checked on our samples after 1-2 days, and no major differences are seen (by eye) in the average size and distribution of condensates. We suspect that the lack of Ostwald ripening effects results from a combination of the relatively large size of condensates (μm scale) (small Laplace pressure) as well as the low solubility of decane in the continuous aqueous phase ($\sim 1 \times 10^{-8}$ v/v).

Here, we provide an estimate of the Ostwald ripening rate, following the analysis in the supporting information of (<https://www.pnas.org/doi/10.1073/pnas.2102014118>). The capillary length of the system l_γ is calculated as follows:

$$l_\gamma = \frac{2\gamma_{ow}}{c_{in}k_bT}$$

Where c_{in} is the number concentration of oil molecules within a condensate, γ_{ow} is the oil-water interfacial tension, k_b is Boltzmann's constant, and T is absolute temperature. The timescale of Ostwald ripening, τ , can then be estimated as:

$$\tau = \frac{R^3 c_{in}}{D l_\gamma c_{eq}}$$

Where c_{eq} is the number concentration of oil molecules in the continuous aqueous phase. We use the following ratio as the volume fraction solubility of decane in the continuous aqueous phase: $c_{eq}/c_{in} = 1 \times 10^{-8}$ (<https://www.sciencedirect.com/science/article/abs/pii/B9780123864543004863>). We approximate the diffusion coefficient of decane in the water to be $D = 1 \times 10^{-10}$ m²/s. We approximate the average radius of the droplets to be 2 μm . This yields $\tau \approx 1.4 \times 10^7$ seconds, or about 1×10^4 days.

lines 199-203: "Semiflexible polymer fibrils such as agarose deform plastically..." - Why are we only now learning this??? This belongs in the introduction.

We thank the Reviewer for pointing this out. We have revised the Introduction section to include basic discussion on the mechanics of fibrillar networks (revised lines 60-64).

- Again, it is unclear how the different mechanical process match up. Here, the authors note that agarose deforms plastically, but then say that this results in bending rigidity dominating. The discussion in lines 120-136 suggests that plastic deformations occur independently of elastic deformations related to the bending rigidity?

We have clarified this in the revised text.

- Which is more relevant to biological cells, the reversible deformation of rubbery gels or the permanent plastic deformation of fibrillar networks?

Fibrillar networks recapitulate much more of the response of intracellular networks. We have included discussion on this in the Discussion (revised lines 378-406).

Surfactant experiments below:

line 208: "excess surfactant" -- In what sense "excess"? I think this only becomes clear later, when surfactant is discussed.

We have clarified this in the revised text (revised lines 315-320). In this particular line, 5% v/v of surfactant is added, well above the critical micelle concentration. The interfacial tension is 0.5 mN/m in this case.

Fig 3d/e: Please check the caption. The decane condensate is red in 2e, not 2d.

We thank the Reviewer for pointing this out, and we have fixed this.

226-227: How does the presence of surfactant "preclude network fracture". Is it just by reducing the strength of capillarity, and hence reducing the incentive for the non-wetting decane to separate from the network? This sentence reads as if the surfactant has some direct chemical influence on the network fibrils themselves, such as increasing their fracture toughness.

Yes, the surfactant simply lowers the oil-water interfacial tension.

228: How do you know whether or not there has been network fracture? Is this just inferred from the shape of the final condensate, or is it visible in the fluorescence imaging?

Fracture is inferred from the condensate shape. We consider samples in which any condensates exhibit a sphericity above 0.95 to be fractured.

lines 230-231: "with no added surfactant, interfacial tension dominates and spherical droplets are exclusively observed (Fig. 4e)." -- In that case, did all of the preceding observations involve some surfactant? Please clarify!

We apologize for the unclear language. No, the only experiments which involved surfactants are Fig. 5b and Fig. 6 in the revised manuscript (Fig. 3e and Fig. 4 previously).

During solvent exchange, the interfacial tension rises with increasing ethanol content. So at early times in all gel concentrations, the condensates have a sufficiently low interfacial tension to permeate throughout the network without fracture. Only later, when the interfacial tension rises sufficiently, do the condensates fracture the network and restructure into spherical droplets. In experiments with enough surfactant added, the interfacial tension never rises sufficiently to fracture the network. We have clarified this in the text (revised lines 100-125).

lines 239-240: "solute is already fully depleted prior to network fracture" -- This is the first reference to solute depletion. Please elaborate, ideally in the introduction during a general overview of the process (see above).

Thank you for pointing this out, we have elaborated on this in the Introduction (revised lines 65-79) as well as first Results section (revised lines 100-125).

Fig 4a: This figure demonstrates some interesting behavior, but suffers from being rather small - it is hard to make out the different morphologies clearly.

In the revised draft, we have sized this figure to an appropriate relative size compared to the other figures to improve visibility.

Reviewer 3

It was a real pleasure to read this manuscript. The experiments are very creative. The phenomena are rather complex, but the authors have done an excellent job of reducing their observations to their key elements and representing the data visually (Fig 2d is gorgeous and clear). The text is excellent – simultaneously compact and lucid.

The problem is fascinating. The competition of condensation, capillarity, and elasticity is unexplored, especially in heterogeneous systems, and the authors make a compelling connection both to recent questions in biophysics as well as more classical phenomena in rigid porous media.

I support publication in Nature Communications provided that the authors can address the following points:

We thank Reviewer 3 for their review of the manuscript and we appreciate the Reviewer's favorable outlook of the manuscript.

1. I am not sure what terminology should be employed to correctly describe their elastic matrix. The authors describe it as fibrillar, but that doesn't seem accurate. To me a fibrillar network would have clearly resolvable linear structures with a persistence length larger than the mesh scale. (like a collagen gel). Here, it looks more like a very heterogeneous molecular gel. Like something that cross-linked while phase separating. In no way does this detract from the significance of the manuscript. The essential point of the manuscript is that they are probing condensation in the limit where the droplets are at a scale that samples network heterogeneity. I think the authors should no longer refer to this as a fibrillar network, and adjust the terminology throughout the rest of the paper accordingly. (NOTE: my understanding is that lyophilized gels tends to look pretty fibrillar in EM even if the swollen networks are not) If the authors insist on fibrillar they will need better in situ structural characterization – maybe with super resolution optical imaging???

In our experiments, we employ agarose as our matrix material. Agarose is a polysaccharide-based semiflexible polymer which is known to form ~10 nm diameter fibrils with a variable mesh size from about 100 nm – 3 um based on gel concentration (<https://pubs.acs.org/doi/10.1021/acsmacrolett.9b00258>). AFM has also been used to characterize agarose networks in the hydrated state and have demonstrated the fibrillar structure of the network (<https://analyticalsciencejournals.onlinelibrary.wiley.com/doi/10.1002/elps.1150180111>).

Based on our understanding, artifacts from freezing hydrogels result from the formation of large ice crystals due to slow cooling during the freezing process. In our experiments, we circumvent this by freezing our agarose gels in liquid-nitrogen-cooled liquid ethane. The much higher thermal conductivity of liquid ethane prevents large ice crystallite formation. This is the same approach used in the preparation of vitreous ice samples for cryo-TEM.

2. Lines 119-121. In the same sentence, the authors refer to the structure being sub-isostatic but having a finite shear modulus. This seems contradictory. Please clarify.

We thank the Reviewer for pointing this out. Agarose gels have a finite shear modulus despite having a sub-isostatic connectivity due to the finite bending modulus of the individual fibrils themselves (<https://pubs.acs.org/doi/10.1021/acsmacrolett.9b00258> and <https://pubs.acs.org/doi/full/10.1021/acs.macromol.0c00601>). We have clarified the language in the revised manuscript (revised lines 179-186).

3. Because of my comments in point 1, I think that the discussion of the yielding of 'fibrils' and its comparison to collagen (lines 130ish) is dodgy and should probably be removed.

Based on our explanation in response to point 1, we do believe that agarose hydrogels have a fibrillar structure. However, we thank the Reviewer for this point; we have removed the comparison to collagen fibrils.

Reviewer 4

The study aims to understand the effects of fibrillar networks on liquid-liquid phase separation occurring in them. The condensates formed inside networks with commensurate mesh-size grow sporadically in bursts. The deformation of the network caused by the formation and growth of condensates is permanent. The dynamics of condensate growth is dependent on the condensate capillarity and network strength. The material of the paper is relevant to the field and is clearly written. But there are some significant concerns regarding the scope, analyses and presentation. The primary novelty of the paper is in studying the sporadic growth dynamics of the condensates in the network. Multiple publications previously have studied condensates inside networks; in fact quite a few of such publications have been cited here. For example, in ref 13 Fig. 4b, and ref 16 shows droplets with similar size and the mesh in actual biological contexts. Ref 22 discusses the physics of droplet formation inside networks and the forces they exert on the fibers. Coupled with these existing concepts, the sporadic growth has the potential to shed light on key aspects of phase separation inside networks. Unfortunately, the manuscript has been presented entirely based on observations without contextualizing them or answering any questions that have been raised in the introduction section as the motivation of the study which fails to communicate the significance of the results. For the above reasons I do not believe this manuscript is ready for publication.

We thank Reviewer 4 for their careful review of the manuscript and for their constructive comments. In response to these comments, we have substantially revised the manuscript. Below, we address the Reviewer's specific points and questions.

Specific concerns: Introduction:

Line 45: Reference 21 is a review paper. The authors should mention specific problems in probing droplets in vivo that their experimental model can address. This is especially necessary since there exists a large body of literature on probing droplets in vitro.

Thank you, we have modified the language here (revised lines 40-43).

Line 52: The wording "continuum-level mechanics are permitted since condensates and molecular strands" is slightly confusing. Maybe the language can be clarified.

We thank the Reviewer for pointing this out. We have now changed the language to clarify the meaning in these lines (revised lines 49-52).

Line 68: Please provide a definition/ description for condensate capillarity.

We have now done this in the Introduction section (revised lines 58-60).

Results:

Overall: Most of the calculations done are based on a single dataset which can be a source for unintended bias. A more statistically robust approach is required.

Since in this manuscript we only intend to introduce the basic mechanics of liquid-liquid phase separation in fibrillar networks, we believe that the use of a reduced number of data sets is sufficient.

Figure 1: Why are all the lobes smaller than the mesh size calculated in Supplementary Text 1? Given the standard deviation shown in Fig. S3, some lobes should be substantially larger. Also, time points should be mentioned in Fig. S4 to avoid confusion. Are the 4 points in Fig. S4A same as Fig.1 b1, b2, b3 and b6? If so, then why does the placement of the red dots corresponding to b2 and b3 seem different in Fig. S4?

Based on this and the other Reviewer's comments, we have remade Fig. 1 to depict fluorescence data as well.

Line 83: Tracking just the lateral area and interpolating the volume can lead to errors. Needs to be addressed. Also, any drift in Z-direction (a common problem with confocal microscopes) can introduce substantial errors in the calculations described in Supplementary Text 2. Given that a confocal microscope is used, Z-stacks should be recorded and the volumes calculated from there.

Based on the Supplementary Movies, we do not see substantial Z-direction drift over the course of the movies. Z-stacks would indeed provide a better measurement of the condensate volumes, however, the time resolution of Z-stack movies (~2-5 minutes per full volume) is insufficient to capture any of the dynamics we describe here.

Line 85: Lobe 1 actually appears around 72s in Supplementary Video 1.

We thank the Reviewer for catching this error; although Fig. 1 and Supplementary Video 1 have been replaced by updated figures and movies.

Line 87: Assuming the flowrates are calculated based on the jumps in Fig. 1c, the second jump clearly is not nearly twice times the size of jump 1. The discrepancy in scaling needs to be addressed.

In response to various comments, we have performed new experiments and recreated Fig. 1.

Line 88: How is the condensation rate of 0.3 fL/s calculated? It is not mentioned in Supplementary Text2.

We have updated the Supplementary Information to include discussion on where the condensation growth rate of is obtained from. Note that the value is slightly updated (0.23 fL/s) (see Supplementary Note 2).

Line 88: What is a capillarity-driven jump? The term needs to be explained adequately before concluding.

We have clarified the definition of a capillarity-driven jump as well as included additional discussion on these jumps (revised lines 146-163).

Line 96: Mentions Supplementary Text 3. The jump in Fig. S5 happens over 40 seconds as opposed to 2. In Supplementary Video S5, the lobe formation happens between 108 and 110 seconds. What is the rationale for choosing 90 and 130s?

We have created a new Fig. 2 which includes specific data showing these capillarity-driven jumps. The new Fig. 2 better demonstrates the kinetics of the abrupt interface jumps.

Line 99: If lobe formation is redistribution, then the overall volume of the condensate should not change. This needs to be shown by comparing actual volumes from z-stacks. Corresponding decrease in thickness of another lobe is not conclusive. Another possibility is that formation of a new lobe deforms the network and generates a strain which causes the shrinking of other lobes.

Z-stacks do not have sufficient time resolution to capture these kinetics. Condensation continues throughout the time in which growth jumps occur. A growth jump occurs over the course of a 2-second image exposure time, but a Z-stack with sufficient resolution and signal takes ~2-5 minutes to acquire.

Line 101: Define/ describe what is an interface jump.

We have clarified the language (revised lines 106-109 and 139-145).

Figure 2: Mention agarose concentration. Fig. 1 is 0.3% and Fig. 3 is a mix of 2%, 0.8% and 0.3%. Also, why is the process shown Fig.2 much slower than in Fig. 1? Between a4 and a5, a lot of structures around the bigger condensate seems to disappear. Are they coalescing? That would be antithetical to the argument made in Lines 197-198.

Thank you for this comment, we have now mentioned the agarose concentration in all experiments. The process shown in the original Fig. 1 depicted growth via abrupt jumps (analogous to Haines jumps). The process shown in the original Fig. 2 (now Fig. 3) depicted the local fracture of the network which allows an already-phase separated condensate to restructure into a spherical droplet. The condensate is originally composed of a continuous, tortuous body which does indeed coalesce into the large spherical droplet when the network is fractured.

Line 123: "Beyond a critical strain..." This critical strain seems to be the most important factor in condensate growth. This strain should correspond to a particular size ratio of the condensate and mesh-size. The strain should be estimated and rearrangement of the network should be studied.

This strain is on the order of magnitude of the mesh size. The network cannot deform or expand much beyond the mesh size due to the formation of closed loops. We have added discussion of this into the revised text (revised lines 190-196).

Line 143: "Subsequent network deformation proceeds via further fracture" is problematic. Why does it not form more lobes? What determines the relative competition between squeezing into a new lobe vs fracturing fibrils to form spherical droplets? For example, in Fig.1 (and Fig. S4) the system keeps adding lobes to the structure whereas in Fig.2 (and Fig. S8) a two-lobe structure fractures the fibrils. This needs to be addressed.

Lobe growth only occurs during phase separation, as oil solute condenses from solution. Once this solute is depleted, then lobe growth stops. However, the interfacial tension of the oil-water interface continues to rise during solvent exchange, which continues to proceed after the oil solute is fully depleted. This increase in interfacial tension is what causes the network to fracture. We have added a new section into the revised text which describes this entire process (revised lines 100-125).

Lines 144-147: The logic here seems inverse. The cavity expansion should be easier (faster) if fracturing strands are easier than dissipating the tension through the network. So fracturing fibrils should become easier with increasing cavity expansion velocity. Also, Fig. S9 does not show increase in velocity in all cases. It shows a drop from 0.3% to 0.8% which is not even discussed. Again, any analysis based on one dataset is not rigorous.

We thank the Reviewer for pointing this out. We originally meant for this line to explain that fracture of the first restraining element is more difficult for higher gel concentrations. However, we recognize the confusion that these lines cause, and we have removed these sections from the revised manuscript.

Lines 168-170: The statement indicates that the expansion of the cavity can be used to probe the microstructure of the network. But no such information is extracted in this work.

We have replaced "probes" with "explores", as that is what we meant.

Figure S8: Red and green panels are a complete mismatch in C and D. The cavity growth in both these cases is uniform and lacks any tortuosity. Also, cavity expansion velocity is calculated from the condensate fluorescence which complicates the problem further.

We have removed these sections as well.

Figure 3: Why is the SEM image from a different cgel from the rest of the main manuscript? Fig. S18 seems to have all concentrations. But the thick shell of fiber seen in 2% (and 1.3%) is lacking in 0.3%.

This needs to be addressed. More importantly, the mesh-sizes from B show that 0.8% has a larger mesh-size than 0.3%. Might be the issue in just one image, but there are no other images to compare with. Why does panel d lack the fluorescence from the decane (as in panel e)? All conditions should be reported uniformly.

We use the SEM image from 2.0% w/w to best visualize the densified shell. The shell does not appear as densified in the lower gel concentrations so it is not easy to visualize in the SEM, although in the confocal it is easier to see. In new Fig. 5a, the time series dissolution experiments use the bright-field images because the dye molecules become solubilized by the surfactant during time-series experiments. In the new 5b, we are able to use fluorescent dyes because we only show before and after images, without the time series.

Lines 230-231: Authors claim that for 0.3% agarose gel and no surfactant case all droplets are spherical. Does that mean droplets showed in Fig.1 will eventually become spherical? If so, what is the timescale? Even in a single condition there seems to be a competition as mentioned in the point for Line 143.

Yes, the droplets in the original Fig. 1 do eventually become spherical. We have updated Fig. 1 to show a full time series of this process, where at later times (approximately $t = 2000$ s), the droplets can be seen becoming spherical.

Discussion:

This is the section that is most concerning as the authors do not 'discuss' their results. There is no discussion as to this sporadic growth model is in agreement or contradiction with other works. If the network fracture eventually creates spherical droplets larger than the mesh size, then what is the significance of the lobe formation? The authors need to discuss specific questions that can be answered from their observations. It is intuitive that condensates inside a network have to nucleate in a size smaller than the mesh-size and grow by fracturing/deforming the network (as discussed in ref. 22). The authors need to justify the novelty of their results, which I assume is the lobe formation. From Fig. 1a(fourth image) it seems that after a certain amount of interface jumps, the network is embedded inside a tortuous condensate which looks strikingly similar to ref 28, which should be brought into the discussion.

We thank the Reviewer for bringing up these point. We have added substantially to the Discussion section in response to these comments (revised lines 378-428).

REVIEWER COMMENTS

Reviewer #1 (Remarks to the Author):

The authors have addressed all the comments I raised. I have no further revisions to request. I believe the MS is ready for publication.

Reviewer #2 (Remarks to the Author):

The authors have addressed many of my previous concerns, but I still have two major concerns and a handful of minor points. Although the authors now distinguish more clearly between fibrillar networks and rubbery gels, I think that many of these distinctions should instead be attributed to the scale of problem (condensate near the mesh size vs much larger than mesh size; see major point 1). The authors have also provided some key additional details about their experimental system that raise several new questions and suggest that Fig 1a provides an incomplete and misleading description of their system (see major point 2). The manuscript is much improved and I am still in favor of publication in Nature Communications, but not until the authors have fully addressed the comments below.

Major points:

1. Phase separation in fibrillar networks vs rubbery gels: My basic understanding is that, in both systems, condensates nucleate below the mesh size and then grow. In both systems, the deformation and fracture of individual network elements will occur as condensates reach and then exceed the mesh size, which is too small to be observable in a rubbery gel. In both systems, bulk mechanics become relevant once condensates greatly exceed the mesh size, which would also eventually happen in the present system if the condensates continued to grow. I'm happy to be convinced otherwise, but the authors did not challenge this view in their response or in the manuscript.

The above may seem like a subtle nuance, but it is crucial for accurately generalising these results and informing this rapidly developing field. The authors present numerous aspects of their observations as being general features of condensate growth in a fibrillar network vs in a rubbery gel, whereas I would argue that many of them are actually general features of the growth of a condensate as it approaches and then exceeds the mesh size vs once it far exceeds the mesh size. The authors must distinguish clearly and explicitly between the former category and the latter, and be honest about where they cannot. It is certainly not the case that condensates will always be commensurate with the mesh size in

fibrillar networks and much larger than the mesh size in rubbery networks -- this is an artifact of experimental design and visualisation constraints. This point is relevant throughout the entire manuscript; some specific examples are listed below.

- lines 54-56: "A condensate growing within [a fibrillar] network experiences no mechanical constraints until it becomes commensurate with the mesh size" -- This statement implies that this is a feature of fibrillar networks but not rubbery gels, which is not true.

- lines 66-68: "By inducing phase separation within fibrillar biopolymer gels where condensates are commensurate with the mesh size, we directly observe the phase separation kinetics and mechanical interactions between condensates and network structural elements." -- Condensates are always commensurate with the mesh size at some point in the process, even in rubbery gels. It would be more clear and accurate to write "We induce phase separation within fibrillar biopolymer gels, where the large mesh size allows for direct observation of..."

- lines 74-76: "The commensurate size of oil condensates and the pore space enables direct mechanical interactions between condensates and network elements" -- This is true, but it will also (presumably) happen in a rubbery gel at very early times. The important feature here is that the mesh size is much larger, so you can see what happens when the condensates reach that size.

- lines 215-223: "Thus, cavity formation is ... can only probe bulk network mechanics." -- Again, I think this is a misleading view. I think the early-stage evolution of condensate growth in a rubbery gel would look very similar to the observations shown here. There is some discussion of connectivity and entropic vs enthalpic forces on lines 181-185, but this is not framed as a comparison between fibrillar vs rubbery networks and it is not clear how this reasoning applies at the mesh scale.

- lines 280-282: "The extent of densification... through the buckling and compaction of fibrils." -- I strongly suspect that this is an artifact of the condensate size relative to the mesh size. In fact, I suspect that it is a generic feature of cavity expansion in an elastic (or elasto-plastic) material, which is a classical problem. The authors should investigate the degree of densification around a cavity in an elastic (or elasto-plastic) continuum as a function of the ratio of the final cavity size to the initial cavity size, the latter being a proxy for the mesh size. It may be that the degree of densification is always much more widespread and pronounced (relative to the cavity size) when this size ratio is small.

- lines 282-283: "This reflects the high compressibility of the fibrillar network: due to the high void fraction of the porous network..." -- I think this is again an artifact of the scale. Rubber networks have similar void fractions. However, I can easily believe that both kinds of networks are much more compressible at the mesh scale than at the continuum scale. Thus, there is an important distinction between mesh-scale mechanics and bulk mechanics that is being blended with the distinction between fibrillar networks and rubbery gels.

- lines 306-310: While polymer gels can sustain high and reversible strains due to rubber elasticity, networks of semiflexible fibrils such as agarose deform plastically upon the application of large strains due to the fibrils' athermal nature. Thus, while the deformation induced by liquid-liquid phase separation in rubbery gels is reversible, we find that phase separation permanently deforms agarose

networks." -- This is an interesting one. Are we certain that there is no near-mesh-scale damage in rubbery networks?

2. Transient evolution: It seems that the evolution of the experimental system is bit more nuanced than was fully communicated in the first version of the manuscript. This does not compromise the results, but it does require more attention and clarity in the revised manuscript. Specifically:

- lines 69-70: "we soak agarose hydrogels in an excess volume of a mixture of decane in ethanol (4% v/v) to diffusively exchange water for the mixture" -- This needs to be stated more clearly.

* What is "an excess volume"? I think the authors mean that the amount of water in the hydrogel is negligible relative to the amount of surrounding ethanol? Please clarify.

* What is the end state of this process? Presumably, the hydrogel ends up being saturated with ethanol/decane rather than water, as illustrated in panel 2 of Fig 1a. Please state.

* Why doesn't phase separation occur during this process, when ethanol/decane diffuses into a water-saturated gel? There will also (transiently) be a mixture of ethanol, decane, and water within the gel.

- lines 70-72: "We induce phase separation by subsequently soaking the mixture-loaded gels in water: as ethanol diffuses out of and water diffuses back into the gel, decane phase separates from the aqueous phase"

* Please state explicitly that the solubility of decane in an ethanol-water mixture decreases strongly with water concentration. This is the driver for phase separation. As noted below, I think a sketch of decane solubility vs time should be included in Fig 1a and a plot of decane solubility vs water (or ethanol) content should be included in the supplementary information.

* As revealed in the response of the authors to my previous point 6 ("What sets the final size of the condensates?"), there is some strong rate-dependence here: "The final size of the condensates depends on the nucleation rate of condensates, which is then set by the solvent exchange rate. The greater the solvent exchange rate (e.g. by using a thinner gel such that solvent exchange via diffusion is faster), the greater the nucleation rate, and in that case, while there are a greater number of condensates, they are on average smaller." This is interesting and important, and should be explained in the manuscript. Also, as noted below, I think Fig 1a needs to include a sketch of the local expected (since it is presumably not directly measurable) water or ethanol concentration as a function of time.

- lines 114-125: "Condensate capillarity is yet insufficient ..." -- I now understand that the interfacial tension is evolving over the course of the experiment, which is an extremely important point! I previously thought that the network eventually fractures as the condensates grow larger and larger, but now it seems that the fracturing is driven by the steady increase in interfacial tension, which continues

long after the condensate has stopped growing (lines 116-125). As noted below, I think Fig 1a needs to explain this process more accurately and clearly, including a sketch of interfacial tension vs time.

- As noted above, Fig 1a is incomplete and/or misleading in a few regards. During panels a2-a5, the system is evolving in a fairly complicated way that is not fully or accurately captured by these cartoons or by the processes labeled (ii), (iii), and (iv).

* The aqueous phase is not pure water in panels a3-a5. The water concentration evolves throughout this sequence.

* Two different processes happen at the same time, both as a result of the evolving water concentration: the decane exsolves and the interfacial tension increases. The former finishes by around 500s (panel a4?) whereas the latter continues for quite a bit longer. Is it fair to say that "(ii) Phase separation" and "(iii) Growth jumps" occur primarily due to exsolution, whereas "(iv) Network deformation" occurs primarily due to increasing interfacial tension after exsolution has finished? The separation of these two timescales (or not) and their separate roles in the overall process need to be clarified.

* Fig 1a should be accompanied by a sketch showing the time evolution of local ethanol concentration, decane solubility, and interfacial tension. This would really help to illustrate what is actually happening in this sequence and to highlight the two different processes and timescales involved. The authors should also include quantitative plots of decane solubility and interfacial tension vs ethanol concentration in the supplementary information.

Minor points:

- lines 23 and 57: "tortuous network pore space"  "network pore space" ? The concept of tortuosity is not explained and does not seem essential or even particularly relevant.

- line 39: "... microtubules^{10,11}, and condensates ..."  "... microtubules^{10,11}. Condensates ..."

- line 58: "interfacial tension"  "capillary forces" to make a concrete link with "condensate capillarity" ?

- lines 61-62: "fibrillar network elements such as those in the cellular interior readily undergo plastic deformation" & line 189: "fibrils begin to yield plastically by bending" -- Do the individual network elements deform plastically, or is it the fibrillar network that deforms plastically via the sliding/rearrangement of fibrils? I thought the latter.

Reviewer #3 (Remarks to the Author):

The authors have adequately address the concerns I raised in my original report. I remain supportive of publication.

Reviewer #4 (Remarks to the Author):

Review of Liu et. al. "Liquid-liquid phase separation within fibrillar networks"

The authors have admirably answered most of the concerns raised by the reviewer. However, there are a few issues that should be addressed.

Main Article:

Figure 2: With the updated figure 1, figure 2 is rather superfluous. Fig. 2c should be combined with Figure 1

Figure 5b: Why is the condensate still tortuous. Based on Figure 6, 0.8% agarose gels should have a spherical condensate.

2nd Response to Reviewers:
“Liquid-liquid phase separation within fibrillar networks”
by Liu et al. for Nat. Comms. NCOMMS-23-06794A

Below, reviewer comments are in blue and our responses are in black.

Reviewer 1

The authors have addressed all the comments I raised. I have no further revisions to request. I believe the MS is ready for publication.

We thank the Reviewer for their review and support of this manuscript.

Reviewer 2

The authors have addressed many of my previous concerns, but I still have two major concerns and a handful of minor points. Although the authors now distinguish more clearly between fibrillar networks and rubbery gels, I think that many of these distinctions should instead be attributed to the scale of problem (condensate near the mesh size vs much larger than mesh size; see major point 1). The authors have also provided some key additional details about their experimental system that raise several new questions and suggest that Fig 1a provides an incomplete and misleading description of their system (see major point 2). The manuscript is much improved and I am still in favor of publication in Nature Communications, but not until the authors have fully addressed the comments below.

We thank the Reviewer for their helpful comments on the revised manuscript.

Major points:

1. Phase separation in fibrillar networks vs rubbery gels: My basic understanding is that, in both systems, condensates nucleate below the mesh size and then grow. In both systems, the deformation and fracture of individual network elements will occur as condensates reach and then exceed the mesh size, which is too small to be observable in a rubbery gel. **In both systems, bulk mechanics become relevant once condensates greatly exceed the mesh size, which would also eventually happen in the present system if the condensates continued to grow.** I'm happy to be convinced otherwise, but the authors did not challenge this view in their response or in the manuscript.

The above may seem like a subtle nuance, but it is crucial for accurately generalising these results and informing this rapidly developing field. **The authors present numerous aspects of their observations as being general features of condensate growth in a fibrillar network vs in a rubbery gel, whereas I would argue that many of them are actually general features of the growth of a condensate as it approaches and then exceeds the mesh size vs once it far exceeds the mesh size.** The authors must distinguish clearly and explicitly between the former category and the latter, and be honest about where they cannot. **It is certainly not the case that condensates will always be commensurate with the mesh size in fibrillar networks and much larger than the mesh size in rubbery networks -- this is an artifact of experimental design and visualisation constraints.** This point is relevant throughout the entire manuscript; some specific examples are listed below.

We thank the Reviewer for bringing up this important point. Indeed in our manuscript we present many of our arguments as a comparison between phase separation in fibrillar networks vs. rubbery gels. Based on our experiments in fibrillar networks and the literature on phase separation in rubbery gels, we believe that the major distinction between these systems is not simply one of condensate size vs. mesh size, but that it is inherent to the mechanics of the material systems themselves. Fibrillar

networks are composed of fibrils (i.e. rods) with a finite bending stiffness and tensile modulus. On the other hand, rubbery gels are composed of crosslinked polymer strands which respond via entropic forces. We think this major distinction is what underlies the many differences which we mention in the manuscript. We respond to specific points below.

- lines 54-56: "A condensate growing within [a fibrillar] network experiences no mechanical constraints until it becomes commensurate with the mesh size"

-- This statement implies that this is a feature of fibrillar networks but not rubbery gels, which is not true.

We thank the Reviewer for pointing this out. Following this comment we have updated the text as such: "A condensate growing within a gel or network experiences no mechanical constraints until it becomes commensurate with the mesh size." (lines 54-55)

- lines 66-68: "By inducing phase separation within fibrillar biopolymer gels where condensates are commensurate with the mesh size, we directly observe the phase separation kinetics and mechanical interactions between condensates and network structural elements."

-- Condensates are always commensurate with the mesh size at some point in the process, even in rubbery gels. It would be more clear and accurate to write "We induce phase separation within fibrillar biopolymer gels, where the large mesh size allows for direct observation of..."

Following this comment we have updated the text as such: "We induce phase separation within fibrillar biopolymer gels, where the large mesh size allows for direct observation of the phase separation kinetics and mechanical interactions between condensates and network structural elements." (lines 66-68)

- lines 74-76: "The commensurate size of oil condensates and the pore space enables direct mechanical interactions between condensates and network elements"

-- This is true, but it will also (presumably) happen in a rubbery gel at very early times. The important feature here is that the mesh size is much larger, so you can see what happens when the condensates reach that size.

We have updated the text to reflect this: "The commensurate size of oil condensates and the pore space enables visualization of direct mechanical interactions between condensates and network elements, and our observations reveal that individual fibrils constrain the shape of condensates." (lines 77-79)

- lines 215-223: "Thus, cavity formation is ... can only probe bulk network mechanics."

-- Again, I think this is a misleading view. I think the early-stage evolution of condensate growth in a rubbery gel would look very similar to the observations shown here.

As we mentioned above, we believe that the major difference between fibrillar networks and rubbery gels is the enthalpic vs. entropic mechanical response of the constituents. In rubbery gels, whose mechanical response largely results from the entropy of the constituent polymer strands, it is a classical result that a cavity will grow unboundedly upon subjection of a driving pressure which exceeds some critical value (that being $p_c = 5E/6$, where E is the gel modulus, for a standard Neo-Hookean material). On the other hand, in a fibrillar network, regardless of the mesh size, the local fracture of restraining fibrils must occur for a cavity to grow since these fibrils cannot stretch elastically like rubbery polymer strands can. And that is the point that we try to convey in those lines. We have updated the text to try and convey this better. (lines 228-235)

There is some discussion of connectivity and entropic vs enthalpic forces on lines 181-185, but this is not framed as a comparison between fibrillar vs rubbery networks and it is not clear how this reasoning applies at the mesh scale.

We realize that the phrasing in the original lines 181-185 is not clear. We did indeed mean for the comparison between enthalpic vs. entropic forces to be between fibrils (enthalpic) and rubbery gels (entropic). This is the updated text: "Unlike the polymer strands in rubbery gels, whose mechanical response largely stems from entropic forces³⁷, the mechanical response of individual semiflexible polymer fibrils such as agarose is largely enthalpic." (lines 193-195)

- lines 280-282: "The extent of densification... through the buckling and compaction of fibrils."
-- I strongly suspect that this is an artifact of the condensate size relative to the mesh size. In fact, I suspect that it is a generic feature of cavity expansion in an elastic (or elasto-plastic) material, which is a classical problem.

Again, the primary distinction we find between cavity growth in rubbery gels (with little network accumulation) versus cavity growth in fibrillar networks (with substantial network accumulation) is that in rubbery gels, network deformation around a growing condensate is largely affine due to rubber elasticity. This means that the strain field propagates far away from the cavity, as molecular network strands far away from the cavity can all deform slightly to accommodate the cavity growth. This mechanism of cavity expansion in a rubbery elastic material is indeed a classical problem.

On the other hand, our experiments demonstrate that deformation of the fibrillar network is highly localized, with only the fibrils in the immediate vicinity of the growing cavity being deformed. Mechanistically, this is very different from the classical problem, and we suspect that this primarily results from the enthalpic mechanical response of the individual fibrils which allows them to bend and buckle.

The authors should investigate the degree of densification around a cavity in an elastic (or elasto-plastic) continuum as a function of the ratio of the final cavity size to the initial cavity size, the latter being a proxy for the mesh size. It may be that the degree of densification is always much more widespread and pronounced (relative to the cavity size) when this size ratio is small.

We thank the Reviewer for this suggestion. This is an area of interest for us, and we are considering an experimental & modeling approach to explore this in future work. In particular, we believe that because of this distinction between deformation in a fibrillar network vs. in a rubbery polymer gel, the use of the compressible Neo-Hookean model in the 1st draft was inappropriate. We are now considering models which explicitly account for the individual fibrils and their mechanics to accurately describe the degree of densification.

- lines 282-283: "This reflects the high compressibility of the fibrillar network: due to the high void fraction of the porous network..."
-- I think this is again an artifact of the scale. Rubbery networks have similar void fractions. However, I can easily believe that both kinds of networks are much more compressible at the mesh scale than at the continuum scale. Thus, there is an important distinction between mesh-scale mechanics and bulk mechanics that is being blended with the distinction between fibrillar networks and rubbery gels.

As we mention, we believe that the primary distinction between cavity growth in fibrillar networks vs. rubbery gels is in their different mechanics. Rubbery networks do indeed have similar void fractions, but network strands do not irreversibly buckle and compact upon cavity growth. Rather, network strands will deform affinely and a strain field will propagate far away from the cavity to accommodate the deformation. On the other hand, fibrillar networks are not able to deform in this way, and rather, deformation is localized immediately to the cavity's edge, thus resulting in a high densification of the network. We thank the Reviewer for this comment, as we have modified the text to clarify the point brought up by the Reviewer, that rubbery networks can have similar void fractions. (lines 294-296)

- lines 306-310: While polymer gels can sustain high and reversible strains due to rubber elasticity, networks of semiflexible fibrils such as agarose deform plastically upon the application of large strains due to the fibrils' athermal nature. Thus, while the deformation induced by liquid-liquid phase separation

in rubbery gels is reversible, we find that phase separation permanently deforms agarose networks." -- This is an interesting one. Are we certain that there is no near-mesh-scale damage in rubbery networks?

There must be some mesh-scale damage occurring in rubbery gels by the same argument as for fibrillar networks: that due to the topology of the network strands (i.e. formation of closed loops), growth of a condensate larger than the mesh size will necessarily fracture some strand loops. However, in rubbery gels, since the mechanical response results from entropic forces, network strands are able to elastically deform to extremely large strains. Because of this rubber elasticity, when the condensates are removed, then the rubbery gel networks can elastically recover their shape and fill up the cavity again, as shown in the various references cited (e.g. <https://doi.org/10.1126/sciadv.aaz0418>). This elastic recovery seems like it should be largely complete even despite some mesh-scale damage in the network.

2. Transient evolution: It seems that the evolution of the experimental system is bit more nuanced than was fully communicated in the first version of the manuscript. This does not compromise the results, but it does require more attention and clarity in the revised manuscript. Specifically:

- lines 69-70: "we soak agarose hydrogels in an excess volume of a mixture of decane in ethanol (4% v/v) to diffusively exchange water for the mixture" -- This needs to be stated more clearly.

* What is "an excess volume"? I think the authors mean that the amount of water in the hydrogel is negligible relative to the amount of surrounding ethanol? Please clarify.

We thank the Reviewer for bringing up this point. We have now clarified the experimental details including processing steps and volumes associated with the gel and various mixtures in the Materials and Methods section (section: Solvent-exchange condensation). In addition, we have made some changes to the wording in the main text to clarify the experimental procedure (lines 68-75).

* What is the end state of this process? Presumably, the hydrogel ends up being saturated with ethanol/decane rather than water, as illustrated in panel 2 of Fig 1a. Please state.

Yes, that is right, we have changed the main text to reflect this (lines 69-71).

* Why doesn't phase separation occur during this process, when ethanol/decane diffuses into a water-saturated gel? There will also (transiently) be a mixture of ethanol, decane, and water within the gel.

In the experimental process, we first soak the hydrogels in pure ethanol to remove the water before soaking the gels in the ethanol/decane mixture. In the previous version of the manuscript, this was only mentioned in the SI. We have now updated the main text to reflect this (lines 68-69).

- lines 70-72: "We induce phase separation by subsequently soaking the mixture-loaded gels in water: as ethanol diffuses out of and water diffuses back into the gel, decane phase separates from the aqueous phase"

* Please state explicitly that the solubility of decane in an ethanol-water mixture decreases strongly with water concentration. This is the driver for phase separation.

We have modified the main text to reflect this statement (lines 72-73).

As noted below, I think a sketch of decane solubility vs time should be included in Fig 1a

We address this point below.

and a plot of decane solubility vs water (or ethanol) content should be included in the supplementary information.

In the new Supplementary Note 1, we have added additional data on the solvent exchange process.

* As revealed in the response of the authors to my previous point 6 ("What sets the final size of the condensates?"), there is some strong rate-dependence here: "The final size of the condensates depends on the nucleation rate of condensates, which is then set by the solvent exchange rate. The greater the solvent exchange rate (e.g. by using a thinner gel such that solvent exchange via diffusion is faster), the greater the nucleation rate, and in that case, while there are a greater number of condensates, they are on average smaller." This is interesting and important, and should be explained in the manuscript.

We thank the Reviewer for this suggestion. However, we are currently studying this phenomenon in further experiments and would not like to discuss it in this manuscript.

Also, as noted below, I think Fig 1a needs to include a sketch of the local expected (since it is presumably not directly measurable) water or ethanol concentration as a function of time.

We address this point below.

- lines 114-125: "Condensate capillarity is yet insufficient ..."

-- I now understand that the interfacial tension is evolving over the course of the experiment, which is an extremely important point! I previously thought that the network eventually fractures as the condensates grow larger and larger, but now it seems that the fracturing is driven by the steady increase in interfacial tension, which continues long after the condensate has stopped growing (lines 116-125).

We address this point below.

As noted below, I think Fig 1a needs to explain this process more accurately and clearly, including a sketch of interfacial tension vs time.

We address this point below.

- As noted above, Fig 1a is incomplete and/or misleading in a few regards. During panels a2-a5, the system is evolving in a fairly complicated way that is not fully or accurately captured by these cartoons or by the processes labeled (ii), (iii), and (iv).

* The aqueous phase is not pure water in panels a3-a5. The water concentration evolves throughout this sequence.

We thank the Reviewer for pointing out that Fig. 1a does not completely capture the system evolution. However, we believe that as a simple cartoon, Fig. 1a sufficiently depicts the processes involved. We only meant for Fig. 1a to schematically depict the phenomena which we show in detail in later figures. We have included additional details in Supplementary Note 1 to elaborate on solvent exchange, phase separation, and temporal evolution of the interfacial tension. Following the Reviewer's comments, we have modified the color palette in Fig. 1a to depict solvent exchange from EtOH to H₂O during steps ii, iii, and iv.

* Two different processes happen at the same time, both as a result of the evolving water concentration: the decane exsolves and the interfacial tension increases. The former finishes by around 500s (panel a4?) whereas the latter continues for quite a bit longer. Is it fair to say that "(ii) Phase separation" and "(iii) Growth jumps" occur primarily due to exsolution, whereas "(iv) Network deformation" occurs primarily due to increasing interfacial tension after exsolution has finished? The separation of these two timescales (or not) and their separate roles in the overall process need to be clarified.

Yes, this is the right reasoning. We thank the Reviewer for bringing up the fact that this was not made entirely clear previously. We have updated the text to make this more clear (lines 132-137)

* Fig 1a should be accompanied by a sketch showing the time evolution of local ethanol concentration, decane solubility, and interfacial tension. This would really help to illustrate what is actually happening in this sequence and to highlight the two different processes and timescales involved. The authors should also include quantitative plots of decane solubility and interfacial tension vs ethanol concentration in the supplementary information.

We address this point above.

Minor points:

- lines 23 and 57: "tortuous network pore space"  "network pore space" ? The concept of tortuosity is not explained and does not seem essential or even particularly relevant.

- line 39: "... microtubules^{10,11}, and condensates ..."  "... microtubules^{10,11}. Condensates ..."

- line 58: "interfacial tension"  "capillary forces" to make a concrete link with "condensate capillarity"?

We thank the Reviewer for the minor comments above and have implemented the suggested changes.

- lines 61-62: "fibrillar network elements such as those in the cellular interior readily undergo plastic deformation" & line 189: "fibrils begin to yield plastically by bending" -- Do the individual network elements deform plastically, or is it the fibrillar network that deforms plastically via the sliding/rearrangement of fibrils? I thought the latter.

Both statements do apply: the individual fibrils can deform plastically by bending and buckling, and the network as a whole can deform plastically by the accumulation of bending and buckling of the individual fibrils.

Reviewer 3

The authors have adequately addressed the concerns I raised in my original report. I remain supportive of publication.

We thank the Reviewer for their review and support of this manuscript.

Reviewer 4

Review of Liu et. al. "Liquid-liquid phase separation within fibrillar networks"

The authors have admirably answered most of the concerns raised by the reviewer. However, there are a few issues that should be addressed.

We thank the Reviewer for their helpful comments on the revised manuscript.

Main Article:

Figure 2: With the updated figure 1, figure 2 is rather superfluous. Fig. 2c should be combined with Figure 1

While Fig. 2 does demonstrate repeated aspects of Fig. 1, we thought it was still important to include Fig. 2 because it more clearly demonstrates the kinetics of the interface jumps, which are central to the manuscript. The experiment in Fig. 2 was performed in a lower concentration gel (larger mesh size) which allowed us to visualize the abrupt lobe shrinkage in Fig. 2c, but this phenomenon is not visible in the experiments for Fig. 1 because the mesh size is smaller.

Figure 5b: Why is the condensate still tortuous. Based on Figure 6, 0.8% agarose gels should have a spherical condensate.

In Fig. 5b, the condensate remains tortuous because we had added surfactant in that experiment to reduce the oil-water interfacial tension. Here is the corresponding text in the caption which describes this: "The surfactant laureth-4 has been added at a concentration of 5% v/v to reduce the oil-water interfacial tension, γ_{ow} , and prevent network fracture."

REVIEWERS' COMMENTS

Reviewer #2 (Remarks to the Author):

The authors have mostly addressed my concerns with their responses and revisions. I thank them for their time and effort. I remain skeptical about a few points, but I think the manuscript is now sufficiently clear that readers can draw their own conclusions. This study will be an excellent addition to the literature on fluid-fluid phase-separation within soft porous materials and I am now happy to recommend it for publication in Nature Communications.

3rd Response to Reviewers:
“Liquid-liquid phase separation within fibrillar networks”
by Liu et al. for Nat. Comms. NCOMMS-23-06794

We thank all Reviewers for their time spent in reviewing this manuscript. The Reviewers' comments were extensive and constructive, and we are very thankful for their contributions.